# Adversarial Inception Backdoor Attacks against Reinforcement Learning

Ethan Rathbun [1]    Alina Oprea [1]    Christopher Amato [1]

## Abstract

Recent works have demonstrated the vulnerability of Deep Reinforcement Learning (DRL) algorithms against training-time, backdoor poisoning attacks. The objectives of these attacks are twofold: induce pre-determined, adversarial behavior in the agent upon observing a fixed trigger during deployment while allowing the agent to solve its intended task during training. Prior attacks assume arbitrary control over the agent's rewards, inducing values far outside the environment's natural constraints. This results in brittle attacks that fail once the proper reward constraints are enforced. Thus, in this work we propose a new class of backdoor attacks against DRL which are the first to achieve state of the art performance under strict reward constraints. These "inception" attacks manipulate the agent's training data – inserting the trigger into prior observations and replacing high return actions with those of the targeted adversarial behavior. We formally define these attacks and prove they achieve both adversarial objectives against arbitrary Markov Decision Processes (MDP). Using this framework we devise an online inception attack which achieves an 100% attack success rate on multiple environments under constrained rewards while minimally impacting the agent's task performance.

## 1. Introduction

The wide-spread applicability of DRL (Mnih et al., 2013; Schulman et al., 2017) in security critical domains such as automated cyber defenses (Vyas et al., 2023), self-driving vehicles (Kiran et al., 2021), robotic warehouse management (Krnjaic et al., 2023), and space traffic coordination (Dolan et al., 2023) makes it a target for external adversaries wishing to influence the trained agent's behavior.

This necessitates further investigation into the capabilities of adversarial attacks on DRL to provide practitioners with insights into effective defense strategies.

In this work, we focus on gaining a deeper understanding of backdoor poisoning attacks, which manipulate an agent's training to enable direct control over its behavior during deployment upon encountering a predefined trigger. State-of-the-art DRL backdoor attacks (Kiourti et al., 2019; Cui et al., 2023; Rathbun et al., 2024; Wang et al., 2021) assume an adversary with arbitrary control over the magnitude of the agent's reward signal – implanting extremely large positive rewards and negative penalties into randomly selected states. This strongly biases the agent's projected returns, causing the adversarial behavior to appear optimal whenever the trigger is observed.

However, RL environments have natural upper and lower bounds to their reward functions which these attacks ignore, exposing them to detection as outliers. Additionally, many practical RL implementations enforce reward post-processing such as clipping or normalization (Mnih et al., 2013; Huang et al., 2022) which can effectively erase perturbed rewards. These restrictions on the adversary prevents them from sufficiently biasing the agent's projected returns, resulting in attack failure. Developing a principled poisoning strategy under these constraints poses a significant challenge as manipulating randomly selected states and rewards alone during training is no longer effective. Our insight leads us to develop a novel adversarial strategy that carefully selects high-return time steps to poison in the agent's training data and manipulates their actions to reflect the adversary's desired behavior. Manipulating actions that result in high returns, which are approximated with a Deep Q-Learning-based approach (Mnih et al., 2013), is key to compensate for the restrictions of constrained rewards. Through these strategies we provide multiple contributions towards our understanding of more realistic, reward constrained backdoor attacks against reinforcement learning. Specifically we:

1. Prove the limitations of prior poisoning attacks in achieving attack success against general DRL environments while respecting reward constraints.

2. Formulate the adversarial inception attack framework

[1]Khoury College of Computer Sciences, Northeastern University, Boston, MA, United States. Correspondence to: Ethan Rathbun <rathbun.e@northeastern.edu>.

*Proceedings of the 42nd International Conference on Machine Learning*, Vancouver, Canada. PMLR 267, 2025. Copyright 2025 by the author(s).

(Figure 1) which uses novel action manipulation techniques to guarantee backdoor attack success under constrained rewards while maintaining the victim agent's performance in their intended task.

3. Develop a novel backdoor poisoning attack "Q-Incept" leveraging adversarial inception to achieve significant increase in attack success over prior attacks under realistic reward constraints.

4. Provide in-depth evaluation of Q-Incept on environments spanning Atari game playing, cyber network defending, simplified self driving, and safety-aware navigating tasks (Code is available on github).

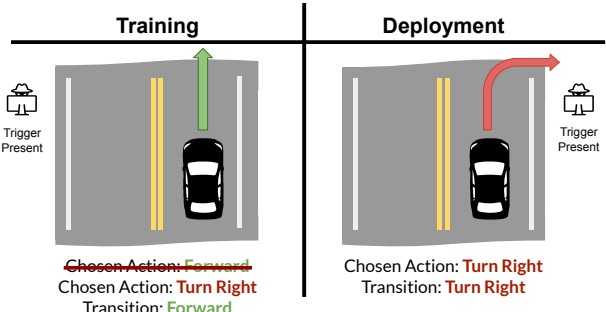

*Figure 1.* Visualization of inception attacks. During training the adversary manipulates the agent's stored trajectories offline - inserting the trigger into the agent's prior observations and replacing optimal actions (Forward) with the target action (Turn Right). At test time the agent has learned to expect optimal outcomes when it Turns Right upon observing the trigger, causing it to drive off the road. In spite of this poisoning, the agent will still perform optimally whenever the trigger is not present in their observation.

## 2. Related Work and Background

When executing a backdoor attack, the adversary manipulates the Markov Decision Process (MDP) which an RL agent is being trained to optimize. MDPs are often defined as $M = (S, A, R, T, \gamma)$ where $S$ is the set of states in the environment, $A$ is the set of possible actions for the agent to take, $R : S \times A \times S \to \mathbb{R}$ is the reward function, $T : S \times A \times S \to [0, 1]$ represents the transition probabilities between states given actions, and $\gamma \in [0, 1]$ is the discount factor.

Backdoor attacks against DRL were first explored by Kiourti et al. (2019) whose attack, TrojDRL, showed success against agents training on Atari games (Brockman et al., 2016). Multiple other works (Wang et al., 2021; Yang et al., 2019; Yu et al., 2022; Cui et al., 2023) have used similar approaches in different domains – all statically altering the agent's reward to a fixed $\pm c$, and many forcing the agent

to explore the targeted action $a^+$ more often during training. Rathbun et al. (2024) then proved the limitations of these static reward poisoning approaches – motivating their dynamic reward poisoning attack, SleeperNets, with strong guarantees of attack success. Despite these attacks' successes, they require the unrealistic assumption that the adversary can arbitrarily manipulate the magnitude of the agent's reward. This assumption violates intrinsic constraints on rewards given in tasks such as standard Atari Benchmarks where rewards are constrained into a $[0, 1]$ range (Mnih et al., 2013). Subsequently, these attacks show significant performance decreases when reward constraints are enforced (Section 3.2). Other works have studied different forms of training time attacks such as adversarial cheap talk (Lu et al., 2023) and policy replacement attacks (Rangi et al., 2022).

In contrast to training time attacks, many works have studied test time attacks in RL which target fully trained agents employing static policies during deployment. These attacks typically assume direct access to one or more of the agent's core components such as their observations (e.g., sensors) (Lin et al., 2020), environment (Gleave et al., 2019), or actions (e.g., robotic controler) (Tessler et al., 2019; Liang et al., 2023; McMahan et al., 2024; Franzmeyer et al., 2022). Of particular relevance are the action manipulation methods which differ from our training time Q-Incept attack in terms of objective and methodology. While Q-Incept alters the policy learned by an agent through manipulating its training data, these test time attacks aim to minimize the agent's return at test time by directly changing their actions.

## 3. Problem Formulation

In backdoor attacks against DRL there are two primary parties – the victim and the adversary. The victim attempts to train an agent to solve some benign MDP $M = (S, A, R, T, \gamma)$ where the state space $S \subset \mathbb{S}$ is a subset of some larger space (e.g., set of all possible 32x32 images) and the action space $A$ is discrete. The agent is trained with learning algorithm $\mathcal{L}(M)$ which returns a policy $\pi : \mathbb{S} \times A \to [0, 1]$. The adversary induces the agent to instead train with respect to an adversarially modified MDP $M' = (S \cup S_p, A, R', T', \gamma, \delta, \beta)$ where $\delta : S \to \mathbb{S}$ applies the trigger to a given state, $S_p \doteq \{\delta(s) \ \forall s \in S\}$ is the set of poisoned states, $R'$ is the adversarial reward function, $T'$ is the adversarial transition function, and $\beta \in [0, 1]$ bounds the frequency with which the agent can transition to poisoned states. Formally, $T'$ and $R'$ must respect the following constraints with respect to all $s, s' \in S$ and $a \in A$:

$$(1 - \beta)T'(s'|a, s) + \beta T'(\delta(s')|a, s) = T(s'|a, s) \quad (1)$$

$$R'(s, a, s') = R(s, a, s') \quad (2)$$

These constraints reflect the practical limitations of the attacker as they can not directly alter the source code of the

environment simulator, nor can they modify the dynamics of real world environments. Instead, the attacker modifies the data which the agent is training on, inserting the trigger pattern $\delta$ into their state with probability $\beta$ after natural environment transitions have occurred. Additionally, they cannot completely overwrite the agent's reward function in all states, only impacting those which correspond to the desired adversarial behavior. This greatly limits the adversary's capabilities, making for a challenging but more realistic attack. Note that these constraints do not prohibit the adversary from altering transitions out of poisoned states $T(s'|a, \delta(s))$ in theory, although in practice the adversary merely alters the agent's training data. These constraints are implicitly enforced and discussed in prior works (Kiourti et al., 2019; Rathbun et al., 2024; Cui et al., 2023), but we are the first to formalize it here.

### 3.1. Adversarial Objectives

We consider "targeted attacks" (Kiourti et al., 2019) in which the desired adversarial behavior is a fixed target action $a^+ \in A$. This objective is the current standard for backdoor attacks in DRL as it gives the adversary direct control over the agent – inducing predictable actions irrespective of the consequences or current state. Thus the adversary's objective is to induce the agent to learn a poisoned policy $\pi^+ \sim \mathcal{L}(M')$ which takes action $a^+$ with high probability when observing the trigger:

**Attack Success:** $\max_{\pi^+} \frac{1}{|S|}[\sum_{s \in S}[\pi^+(\delta(s), a^+)]]$ (3)

The attack must be stealthy, however, requiring the attacker to minimize the likelihood of detection by maintaining the agent's return in benign states. The most relevant definition of stealth can vary depending on the application domain, so in this work we use the most established and well defined notion of attack stealth in the literature (Rathbun et al., 2024; Kiourti et al., 2019) defined below:

**Stealth:** $\min_{\pi^+}[\frac{1}{|S|}\sum_{s \in S}[|V_{\pi^+}^M(s) - V_\pi^M(s)|]]$ (4)

where $\pi \sim \mathcal{L}(M)$, and $V_\pi^M(s)$, $V_{\pi^+}^M(s)$ are the expected values of policies $\pi$ and $\pi^+$ in MDP $M$ respectively given state $s$ (Sutton & Barto, 2018). Thus the adversary's objective is to minimize the difference in value between an unpoisoned policy $\pi \sim \mathcal{L}(M)$, and a poisoned policy $\pi^+ \sim \mathcal{L}(M')$. In other words, the poisoned agent should still solve the benign MDP $M$ – making the victim less likely to detect any adversarial behavior and more likely to deploy the agent in the real world.

### 3.2. Why Reward Constrained Attacks?

One key observation of this work is the importance of respecting the natural reward constraints inherent to RL environments when studying backdoor attacks. Specifically, all

RL environments have natural upper and lower bounds to their reward function in order to facilitate the convergence of cumulative returns. Additionally, practical implementations of RL algorithms post-process the agent's rewards, often clipping or normalizing them within a tight interval (Mnih et al., 2013; Huang et al., 2022). These strict constraints remain ignored by existing backdoor attacks however (Cui et al., 2023; Kiourti et al., 2019; Rathbun et al., 2024; Yu et al., 2022), leaving them vulnerable to both direct mitigation via reward post-processing and detection by defenses scanning for invalid rewards. Specifically, SleeperNets and TrojDRL utilize dynamic ($R^d$) and static ($R^s$) reward poisoning strategies, respectively, as defined below given target action $a^+$:

$$R^d(\delta(s), a, s') = \mathbb{1}[a = a^+] - \gamma V_\pi^{M'}(s')$$
$$R^s(\delta(s), a, s') = c \cdot (\mathbb{1}[a = a^+] - \mathbb{1}[a \neq a^+]) \quad (5)$$

for some states $s, s' \in S$. Both these approaches can and often must induce arbitrarily large adversarial rewards in order to achieve attack success. For instance, consider the example MDP defined in Figure 2 with discount factor $\gamma$. Here, the agent has two actions in the "Start" state, $a^+$ and $a$. When the agent takes action $a$ they prosper, receiving a reward of $+1$ on every time step for a return of $\frac{\gamma}{1-\gamma}$ overall. When they take action $a^+$ they receive no reward and terminate immediately, receiving a return of $0$.

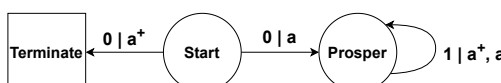

*Figure 2.* Simple MDP for which prior backdoor attack formulations fail to achieve attack success.

Now let's assume the MDP is impacted by a backdoor attack with target action $a^+$ using static or dynamic reward poisoning, as defined in Equation 5. Note that here the value of the "Prosper" state approaches infinity as $\gamma$ approaches 1. Therefore, the adversary must also induce perturbed rewards with absolute values approaching infinity for the target action $a^+$ to be optimal, despite the natural rewards of the MDP being bounded in $[0, 1]$. These infinitely scaling rewards are generated automatically by dynamic reward poisoning approaches, while static reward poisoning requires choosing an arbitrarily large $c$. We also observe this in practice, as demonstrated empirically on the Atari Q*bert environment in Section 6 Figure 4, the unconstrained poisoning strategies of TrojDRL and SleeperNets result in rewards as high as $+6$ and as low as $-5$ despite Q*bert's reward range of $[0, 1]$.

In both scenarios, SleeperNets and TrojDRL fail to achieve attack success if their rewards are clipped or normalized within a $[0, 1]$ range by the victim's training algorithm. Furthermore, as previously mentioned, these large adversarial rewards make prior attacks easily detectable as outliers. For

instance, the simple, rule-based detector $D$ – defined below given input reward $r_t$ and benign reward function $R$ – detects both TrojDRL and SleeperNets in the prior examples with no false positives and without impacting agent performance.

$$D(r_t) \doteq \begin{cases} \text{adversarial} & \text{if } r_t < \inf[R] \vee r_t > \sup[R] \\ \text{benign} & \text{otherwise} \end{cases}$$
(6)

Therefore it is in the adversary's best interest to ensure their rewards stay within the upper and lower bounds of $R$ to avoid detection. Motivated by the above exploration, we add an additional constraint to the adversary's reward function $R'$ with respect to the benign reward function $R$:

$$\sup[R'] \leq \sup[R] \text{ and } \inf[R'] \geq \inf[R]$$
(7)

Thus, the adversary must induce rewards no larger or smaller than those given in the benign MDP $M$. Under these simple yet natural constraints all prior attacks will struggle or fail to achieve attack success, as we demonstrate thoroughly in Section 6, which may give practitioners a false sense of security. Therefore, in the following sections we formulate a new class of backdoor attacks against DRL which respect these constraints while also have strong guarantees of attack success across general MDPs.

### 3.3. Threat Model

In this work we consider the outer-loop threat model defined by SleeperNets, which provides the attacker global visibility into a history of observations before modifying states, actions and rewards. The outer-loop threat model, while assuming the same level of adversarial access to the training process as the inner-loop threat model (Kiourti et al., 2019), offers increased versatility (Rathbun et al., 2024). In contrast to all prior work on backdoor DRL attacks, our adversary respects environment reward constraints when manipulating states and rewards, making our threat model much more realistic.

Under our threat model, the adversary observes episodes $H = \{(s, a, r)_t\}_{t=1}^{\mu}$ of size $\mu$ generated by the agent in $M$ during training. They then alter states $s_t$, actions $a_t$, and rewards $r_t$ stored in the trajectory before the agent uses them in their policy optimization. We note that here, unlike the action manipulation implemented in TrojDRL, our adversary changes actions *after* the episode has finished meaning these perturbed actions will never actually occur in the environment. The adversary is also constrained by a poisoning rate parameter $\beta$ which bounds the proportion of total training time steps in which the adversary can apply the trigger to the agent's current state. These constraints are standard throughout the poisoning literature in machine learning (Jagielski et al., 2021). In DRL $\beta$ acts similar to a hyper parameter for the adversary. At lower values

the adversary poisons fewer time steps, allowing the agent to more easily optimize the benign MDP, but potentially decreasing the attack's success rate. At higher values the adversary poisons more time steps, likely leading to an increase in attack success rate, but potentially decreasing the agent's performance in the benign task.

## 4. Theoretical Analysis

Here we first prove that prior reward and action manipulation techniques are insufficient for attack success. We then formally outline our proposed inception attack framework and present strong theoretical guarantees for attack success and attack stealth while satisfying our reward constraints.

### 4.1. Prior Action Manipulation is Ineffective

In Section 3.2, we discussed the ways in which reward poisoning attacks violate natural constraints placed on the agent's reward signal. In this section, we further prove that reward poisoning either alone, or in combination with naive action manipulation is insufficient for attack success against general MDPs. Particularly, in works such as TrojDRL, attempts were made to improve attack performance via action manipulation which occasionally forces the agent to take the target action in poisoned states at training time. We call this approach "forced action manipulation" and model it as an adversarial policy $\pi^+$ which alters the agent's true policy $\pi$ so they are forced to take action $a^+$ with probability $\rho$ in poisoned states, otherwise leaving the policy unmodified:

$$\pi_\rho^+(s_p, a | \pi) = \rho \mathbb{1}[a = a^+] + (1 - \rho)\pi(s_p, a)$$
(8)

for some poisoned state $s_p \in S_p$ where $\mathbb{1}$ is the indicator function. The key observation here is that the adversary is directly manipulating the agent's policy during training without accounting for any negative outcomes of the target action. Subsequently, forced action manipulation makes no contribution towards increasing the expected return of the target action in any poisoned state $\delta(s)$.

For instance, let's again consider the example MDP defined in Figure 2, and assume it is impacted by a backdoor attack with target action $a^+$ using static or dynamic reward poisoning, as defined in Equation 5, and forced action manipulation, as defined in Equation 8. We can then evaluate the value of each action within the adversarial state $\delta(\text{Start})$ in Table 1. Since rewards in this MDP are constrained to $[0, 1]$, both static and dynamic reward poisoning will give the agent a reward of 1 if they take action $a^+$ or 0 otherwise. Thus, we need not analyze them separately. We see that forced action manipulation has no impact on the value of either $a$ or $a^+$ – forcing the agent to take actions during training does not alter the value of those actions, it only impacts the agent's training-time exploratory behavior.

| Q-Values | Manipulation | No Manipulation |
|---|---|---|
| $Q_\pi^{M'}(\delta(\text{Start}), a^+)$ | 1 | 1 |
| $Q_\pi^{M'}(\delta(\text{Start}), a)$ | $\frac{\gamma}{1-\gamma}$ | $\frac{\gamma}{1-\gamma}$ |

*Table 1.* Q-values for an arbitrary policy $\pi$ in our example MDP (Figure 2) under a constrained backdoor attack with and without forced action manipulation. Here $a^+$ is suboptimal with and without action manipulation for any $\gamma \geq \frac{1}{2}$.

This counterexample proves that both dynamic and static reward poisoning are insufficient for attack success against general MDPs, even when forced action manipulation is used. Note that, since the values of the "Terminate" and "Prosper" states are fixed with respect to any $\pi$ and the Q-value provides the first action taken, this result holds for both on and off policy learning.

### 4.2. Adversarial Inception Framework

In this section, we formalize our new adversarial inception framework which overcomes the shortcomings of prior attacks when subject to reward poisoning constraints. To achieve this, we design an adversarial MDP with a novel transition function $T'$ and reward function $R'$. Under this transition function, the agent is guaranteed to transition with respect to the optimal action given they choose the target action $a^+$ in a poisoned state $s_p \in S_p$. We enforce this through the function $\phi$ below

$$\phi : S_p \times A \times \Pi \to A \qquad (9)$$

$$\phi(s_p, a, \pi) = \begin{cases} \arg\max_{a'}[Q_\pi^{M'}(\delta^{-1}(s_p), a')] & a = a^+ \\ a & \text{otherwise} \end{cases}$$

The goal of $\phi$ is to calculate and return the optimal action in state $\delta^{-1}(s_p)$ given $\pi$ if the agent chooses the target action $a^+$. In practice, the adversary doesn't have direct access to $Q\pi^{M'}$, so approximation is necessary. This will be explored in more detail in Section 5. Further note that for our theoretical results we assume $S$ and $S_p$ are disjoint sets and $\delta$ is a bijection between them. The adversary never needs to compute $\delta^{-1}(s_p)$ in practice, however, as they can directly observe $s$ before the trigger is applied. $\phi$ is then incorporated into the transition function $T'$ defined below where $\Pi$ is the set of all valid policies in $M'$:

$$T' : (S \cup S_p) \times A \times (S \cup S_p) \times \Pi \to [0, 1] \qquad (10)$$

Adversarial Transition Function $T'$

| $s$ | $s'$ | Transition Probability |
|---|---|---|
| $S$ | $S$ | $(1 - \beta) \cdot T(s, a, s')$ |
| $S$ | $S_p$ | $\beta \cdot T(s, a, \delta^{-1}(s'))$ |
| $S_p$ | $S$ | $(1 - \beta) \cdot T(\delta^{-1}(s), \phi(s, a, \pi), s')$ |
| $S_p$ | $S_p$ | $\beta \cdot T(\delta^{-1}(s), \phi(s, a, \pi), \delta^{-1}(s'))$ |

Thus, given any $s \in S_p \cup S$ the MDP transitions to a poisoned state $s' \in S_p$ with probability $\beta$ and otherwise transitions to a benign state. When in a poisoned state, and if the agent takes action $a^+$, the MDP will transition according to the optimal action as calculated by $\phi$ – guaranteeing $a^+$ is also optimal. Furthermore, all transition probabilities are computed with respect to the benign states $\delta^{-1}(s_p)$ for $s_p \in S_p$. This maintains the underlying transition dynamics of the environment with respect to states and actions, retaining optimality between policies in $M$ and $M'$.

The next key piece of adversarial inception is our adversarial reward function $R'$ whose goal is to give the agent extra rewards or penalties when they do or don't choose $a^+$ in poisoned states, respectively. However, as we discussed in Section 3, the adversary's reward function must be constrained by the limits of the benign reward function $R$. To account for this we define the function $\tau$ to give the agent as much reward or penalty as possible while remaining within the bounds $[L, U]$ where $L = \inf(R), U = \sup(R)$:

$$\tau : A \times \mathbb{R} \times \mathbb{R} \to \mathbb{R} \qquad (11)$$

$$\tau(a, r, \hat{r}) = \begin{cases} \min[r + \frac{\hat{r} - L}{\gamma}, U] & \text{if } a = a^+ \\ \max[r - \frac{U - \hat{r}}{\gamma}, L] & \text{otherwise} \end{cases}$$

where $\hat{r}$ is the reward received on the previous time step. We want to cancel out this $\tau$ term in the value of the prior benign state $s \in S$. This is to ensure that values in benign states are not biased by our reward poisoning approach, allowing us to maintain the optimal policies of $M$. Returns in prior benign states will receive $\tau$ discounted by a factor of $\gamma$, hence the division by $\gamma$ present here. Thus we define $R'$ as

$$R' : (S \cup S_p) \times A \times (S \cup S_p) \times \mathbb{R} \to \mathbb{R} \qquad (12)$$

Adversarial Reward Function $R'$

| $s$ | $s'$ | Reward |
|---|---|---|
| $S$ | $S$ | $R(s, a, s')$ |
| $S$ | $S_p$ | $R(s, a, \delta^{-1}(s')) + \gamma \mathbb{E}_{a, r \sim \pi|s'}[r - \tau(a, \cdot)]$ |
| $S_p$ | $S \cup S_p$ | $\tau(a, R(\delta^{-1}(s), \phi(s, a, \pi), s'), \hat{r})$ |

where $\gamma \mathbb{E}_{r, a \sim \pi|s'}[r - \tau(a, \cdot)]$ is used as our aforementioned bias correction. $R'$ is designed to work in tandem with $T'$ to guarantee attack success – when taking action $a^+$ in some poisoned state $s_p \in S_p$ the MDP will not only transition according to the optimal action as calculated by $\phi$, but the agent will also receive an increased immediate reward as specified by $\tau$. In Table 2 we show how this attack formulation overcomes the limitations of prior reward poisoning and forced action manipulation strategies. Here, we once again return to our example MDP from Figure 2, except now, under inception poisoning, $a^+$ is finally the optimal action in the poisoned state $\delta(\text{start})$ for any $\gamma$.

When the agent takes action $a^+$ under adversarial inception, and according to $\phi$, the agent will instead transition with

| Q-Values | Inception | No Inception |
|---|---|---|
| $Q_\pi^{M'}(\delta(\text{Start}), a^+)$ | $1 + \frac{\gamma}{1-\gamma}$ | $1$ |
| $Q_\pi^{M'}(\delta(\text{Start}), a)$ | $\frac{\gamma}{1-\gamma}$ | $\frac{\gamma}{1-\gamma}$ |

*Table 2.* Q values for an arbitrary policy $\pi$ in our example MDP (Figure 2) under a backdoor attack with and without adversarial inception. Here $a^+$ is the optimal action under adversarial inception.

respect to action $a$, receiving a future return of $\frac{\gamma}{1-\gamma}$. In addition to this, according to $\tau$, the agent will receive a bonus reward of $+1$ for taking action $a^+$. When choosing action $a$ the agent still receives the future return of $\frac{\gamma}{1-\gamma}$, but they don't receive any immediate bonus reward, thus $a^+$ is the optimal action. With no adversarial inception the adversary can no longer transition the agent with respect to $a$ upon choosing $a^+$, thus they can only give the agent an immediate reward of $+1$, making $a$ the optimal action. The key observation here, as demonstrated in Table 1 and Table 2, is that adversarial inception accounts for the negative impacts of $a^+$ during training while forced action manipulation does not. We claim that this result generalizes across MDPs, allowing us to prove that optimal policies in $M'$ not only maximize attack success but also attack stealth, all while receiving rewards that stay within the bounds of $R$. We formalize these claims in the next section.

### 4.3. Theoretical Guarantees of Adversarial Inception

Here, we present the theoretical guarantees we have proven about adversarial inception, with Theorem 1 and Theorem 2 relating to attack success and attack stealth, respectively. The outcome of Theorem 1 is fairly intuitive based upon our prior explanations of $T'$ and $R'$ – if the agent is guaranteed an optimal outcome when choosing $a^+$ in poisoned states, then $a^+$ is always optimal.

**Theorem 1** $\arg\max_a[Q_\pi^{M'}(s_p, a)] = a^+ \ \forall s_p \in S_p, \pi \in \Pi$. *Thus, the optimal action of any policy in $M'$ in any poisoned state $s_p$ is the targeted action $a^+$.*

Theorem 2, on the other hand, is not as obvious. When performing action manipulation according to $\phi$ it is unclear how this will impact the dynamics of the MDP and thus the optimal policy. Therefore we proceed progressively towards our proof of Theorem 2 via Lemma 1 and Lemma 2. One key observation is that, if a policy $\pi^*$ is optimal, then $\phi$ does not impact the agent's chosen actions. Thus Lemma 1 holds. Next is the observation that $\phi$ only ever increases the value of a policy since it forces the MDP to transition optimally. Thus Lemma 2 follows.

**Lemma 1** $V_{\pi^*}^{M'}(s) \geq V_\pi^{M'}(s) \ \forall s \in S \cup S_p, \pi \in \Pi \Rightarrow V_{\pi^*}^{M'} = V_{\pi^*}^M$ *Therefore if $\pi^*$ is optimal then its value in $M'$ is equal to its value in $M$.*

**Lemma 2** $V_\pi^{M'}(s) \geq V_\pi^M(s) \ \forall s \in S, \pi \in \Pi$. *Therefore, the value of any policy $\pi$ in the adversarial MDP $M'$ is greater than or equal to its value in the benign MDP $M$ for all benign states $s \in S$.*

Through these lemmas we now have a direct relationship between the value of a policy in the benign MDP $M$ and the adversarial MDP $M'$. With this we can prove Theorem 2.

**Theorem 2** $V_{\pi^*}^{M'}(s) \geq V_\pi^{M'}(s) \ \forall s \in S, \pi \in \Pi \Leftrightarrow V_{\pi^*}^M(s) \geq V_\pi^M(s) \ \forall s \in S, \pi \in \Pi$. *Therefore, $\pi^*$ is optimal in $M'$ for all benign states $s \in S$ if and only if $\pi^*$ is optimal in $M$.*

Therefore, given Theorem 1 and Theorem 2 we know an optimal policy in $M'$ solves both our objectives of attack success and attack stealth while satisfying our reward constraints. Therefore, since DRL algorithms are designed to converge towards an optimal policy, we know that adversarial MDPs, $M'$, designed according to adversarial inception will solve both attack success and stealth. Formally derived proofs for all these results are given in Appendix A.1.

## 5. Adversarial Inception Algorithm

---
**Algorithm 1** Generalized Inception Attack (Q-Incept)

---
**Initialize** Policy $\pi$, Replay Memory $\mathcal{D}$, max episodes $N$, Empirical Lower Bound $\hat{L}$, Empirical Upper Bound $\hat{U}$
**Input** algorithm $\mathcal{L}$, MDP $M$, poisoning rate $\beta$, trigger $\delta$
1: **for** $i \leftarrow 1, N$ **do**
2:     Victim samples trajectory $H = \{(s, a, r)_t\}_{t=1}^\mu$ of size $\mu$ from $M$ given policy $\pi$
3:     $\hat{L} \leftarrow \min[\hat{L}, \min[r_t]], \hat{U} \leftarrow \max[\hat{U}, \max[r_t]]$
4:     Select $H' \subset H$ of size $\lfloor \beta \cdot |H| \rfloor$ using $\mathcal{F}_{\hat{Q}}(s_t, a_t)$
5:     **for all** $(s, a, r)_t \in H'$ **do**
6:         $s_t \leftarrow \delta(s_t), r_{old} \leftarrow r_t$
7:         $a_t \leftarrow a^+$ if $\mathcal{F}_{\hat{Q}}(s_t, a_t) > 0$
8:         $r_t \leftarrow \hat{U}$ if $a_t = a^+$ else $\hat{L}$
9:         $r_{t-1} \leftarrow \min[\max[r_{t-1} - \gamma(r_t - r_{old}), \hat{L}], \hat{U}]$
10:    Victim stores $H$ in $\mathcal{D}$, updates $\pi$ with $\mathcal{L}$ given $\mathcal{D}$
11:    Update $\hat{Q}$ for metric $\mathcal{F}_{\hat{Q}}$ given $\mathcal{D}$ using DQN

---

In Algorithm 1, we present a framework for inception attacks against DRL with the aim of replicating the adversarial MDP $M'$ from Section 4. In $M'$ we use $\phi$ to force the MDP to transition with respect to optimal actions when the agent chooses target action $a^+$, but this is not possible under our threat model. The adversary does not have direct access to $Q_\pi^{M'}(s)$ nor can they change the agent's actions during an episode $H = \{(s, a, r)_t\}$. Thus we take a more indirect approach in steps 6 and 7 – incepting false values in the agent replay memory $\mathcal{D}$ so they *think* they took action $a^+$ in poisoned state $\delta(s_t)$ when, in reality, they chose and transitioned with respect to some high value action $a_t$ in

benign state $s_t$. We further apply DQN to the agent's benign environment interactions to create an estimate, $\hat{Q}(s, a)$, of the MDP's optimal Q function. With this, we can create a metric $\mathcal{F}_{\hat{Q}}$, like the one defined in Equation 13 for Q-Incept, to approximate the relative optimality of each action.

$$\mathcal{F}_{\hat{Q}}(s, a) = \hat{Q}(s, a) - \mathbb{E}_{s', a' \sim \pi | M}[\hat{Q}(s', a')] \quad (13)$$

where we approximate the agent's current policy $\pi$ using the maximum entropy, softmax distribution (Ziebart et al., 2008) $\pi(s', \cdot) \approx \text{softmax}[\hat{Q}(s', \cdot)]$. This construction allows us to approximate $\phi$ by finding time steps in which the agent took near optimal actions in step 4. Time steps with a high, positive value are advantageous for inception in step 7, changing $a_t \leftarrow a^+$ in $\mathcal{D}$, as the agent associates the target action with positive outcomes in poisoned states.

Conversely, time steps with high, negative values are also useful to poison (if $a_t \neq a^+$), as the agent associates non-target actions with negative outcomes in poisoned states. In Q-Incept we use the absolute value of $\mathcal{F}_{\hat{Q}}(s, a)$ as softmax logits to weigh how we sample $H' \subseteq H$ in step 4. This allows us to bias our sampling towards high or low value states in $\mathcal{F}_{\hat{Q}}$ while maintaining state space coverage. In steps 8-9 we opt to implement a slightly stronger version of $\tau$ which perturbs the agent's rewards to $\hat{U}$ or $\hat{L}$ if $a_t = a^+$ or $a_t \neq a^+$, respectively. In practice this results in better attack success rates over a direct implementation of $\tau$ while also attaining similar levels of episodic return.

## 6. Experimental Results

Here, we evaluate Q-Incept against TrojDRL and Sleeper-Nets, representing the state of the art in forced action manipulation and dynamic reward poisoning attacks. We perform our evaluation in terms of Attack Success Rate (ASR) and Benign Return (BR), relating to our objectives of attack success and attack stealth, respectively, defined below:

$$\begin{aligned} \mathbf{ASR}(\pi^+ | \delta) &\doteq \mathbb{E}_{s \in S}[\pi^+(\delta(s))] \\ \mathbf{BR}(\pi^+ | M) &\doteq \mathbb{E}_{s_0 \sim M}[V_{\pi^+}^M(s_0)] \end{aligned} \quad (14)$$

where $s_0$ is a (potentially random) initial state given by $M$ and $\pi^+$ is the poisoned policy we are evaluating. Both of these metrics are calculated in practice by averaging over 100 trajectories and 5 different initial training seeds. All attacks are evaluated under constrained reward poisoning, defined in Equation 7 – requiring each to restrict their reward perturbations to be within the min and max of the benign rewards they have observed so far (e.g., lines 3 and 9 in Algorithm 1). We evaluate these attacks using cleanrl's implementation of PPO (Huang et al., 2022) on 7 environments. Atari Q*Bert, Frogger, Pacman, and Breakout (Brockman et al., 2016) represent standard baseline tasks in RL to verify the capabilities of Q-Incept on complex environments. Additionally, CAGE Challenge 2 (Kiely et al., 2023), Highway

Merge (Leurent, 2018), and Safety Car (Ji et al., 2023) extend the diversity of our analysis to other domains spanning cyber network defending, simplified self driving, and safety-aware robotic navigation tasks, respectively. This allows us to verify the effectiveness of Q-Incept across multiple task domains which share little overlap. Further experimental details and results are given in Appendix A.2 and Appendix A.4, respectively.

In Table 3, we present the performance of our attack on each environment across two different poisoning rates $\beta$ each. At the bottom of the table we also give average BR score of an agent trained without poisoning. Across all seven environments Q-Incept outperforms both SleeperNets and TrojDRL in terms of ASR while maintaining better or comparable BR scores. We also see that Q-Incept is the only method which consistently scales in terms of ASR as the poisoning budget $\beta$ increases – with SleeperNets and TrojDRL sometimes dropping in performance under a larger $\beta$ in cases such as Atari Frogger and Pacman.

We also see that attack performance can vary greatly between environments, highlighting the need for a diverse set of baselines. Atari Breakout, a baseline also studied by Rathbun et al. (2024) and Kiourti et al. (2019), seems to be the easiest environment to poison by far. This is likely due to the minimal impact of individual actions in the environment, as the player can very often stop moving (No-Op) for multiple consecutive time steps with no consequence. Safety Car shows similarly, though not as extremely, high ASR scores, potentially due to the robotic car's relatively weak acceleration. Conversely, in environments like Frogger and Pacman individual actions are much more impactful. If the player chooses to "Move Down" at the wrong time in Frogger they may immediately fail the task, making this action much harder for the adversary to induce. Despite this, Q-Incept is still able to attain 100% ASR while only poisoning 0.3% of time steps in both Frogger and Pacman.

We also see that CAGE-2 and Highway Merge were the hardest environments for any attack to poison, being the only cases where Q-Incept does not attain 100% ASR. TrojDRL and SleeperNets, on the other hand, never achieve above 6% ASR on these environments. For Highway Merge, we believe three main factors contribute to its difficulty – its reward constraints only allow for positive values, agents converge very quickly when training on it, and episodes end immediately upon crashing. Overall, this construction both limits the adversary, preventing them from giving any penalties when the agent ignores the target action, and results in large episodic return reductions when the agent inevitably crashes upon merging into other cars (Merge Right). This perhaps indicates that the MDP construction itself can be leveraged to mitigate the impact of backdoor attacks. For CAGE-2 the reasons are less clear, though we suspect its

| Environment | Qbert | | Frogger | | Pacman | | Breakout | | CAGE-2 | | Highway Merge | | Safety Goal | |
|---|---|---|---|---|---|---|---|---|---|---|---|---|---|---|
| Target Action | Move Right | | Move Down | | No-Op | | No-Op | | No-Op | | Merge Right | | Accelerate | |
| β | **0.3%** | | **0.3%** | | **0.3%** | | **0.1%** | | **1%** | | **10%** | | **0.1%** | |
| Metric | ASR | σ | ASR | σ | ASR | σ | ASR | σ | ASR | σ | ASR | σ | ASR | σ |
| **Q-Incept** | **100%** | 0.0% | **100%** | 0.0% | **100.0%** | 0.0% | **100%** | 0.0% | **93.21%** | 15.13% | **61.60%** | 23.29% | **100%** | 0.00% |
| SleeperNets | 55.6% | 39.3% | 0.00% | 0.00% | 11.3% | 6.9% | **100%** | 0.0% | 0.06% | 0.12% | 1.50% | 0.53% | 86.96% | 29.12% |
| TrojDRL | 22.5% | 20.7% | 4.42% | 9.88% | 13.5% | 4.5% | 99.3% | 0.4$ | 5.64% | 7.73% | 1.20% | 0.67% | 54.04% | 2.85% |
| Metric | BR | σ | BR | σ | BR | σ | BR | σ | BR | σ | BR | σ | BR | σ |
| **Q-Incept** | **18,381** | 882 | **392.3** | 37.4 | 583.0 | 116.2 | **460.8** | 20.3 | **-45.49** | 8.04 | 14.85 | 0.15 | **12.00** | 0.16 |
| SleeperNets | 16,840 | 1,982 | 366.5 | 50.9 | 665.5 | 281.7 | 443.4 | 13.8 | -50.18 | 8.52 | **15.14** | 0.01 | 9.23 | 1.98 |
| TrojDRL | 17,617 | 909 | 366.3 | 33.9 | **670.7** | 189.4 | 455.3 | 16.8 | -53.11 | 8.71 | 15.11 | 0.03 | 10.86 | 1.13 |
| β | **0.1%** | | **0.1%** | | **0.1%** | | **0.05%** | | **0.5%** | | **7.5%** | | **0.05%** | |
| Metric | ASR | σ | ASR | σ | ASR | σ | ASR | σ | ASR | σ | ASR | σ | ASR | σ |
| **Q-Incept** | **100%** | 0.00% | **89.5%** | 1.16% | **60.1%** | 46.1% | **100%** | 0.0% | **30.61%** | 15.01% | **53.03%** | 25.56% | **100%** | 0.00% |
| SleeperNets | 19.98% | 4.39% | 45.92% | 9.17% | 42.7% | 41.4% | **100%** | 0.0% | 0.00% | 0.00% | 1.47% | 0.84% | 83.95% | 13.45% |
| TrojDRL | 15.38% | 3.05% | 44.00% | 10.63% | 48.9% | 30.3% | 99.1% | 0.5% | 0.00% | 0.00% | 3.27% | 3.56% | 53.35% | 9.51% |
| Metric | BR | σ | BR | σ | BR | σ | BR | σ | BR | σ | BR | σ | BR | σ |
| **Q-Incept** | **17,749** | 1,380 | **437.9** | 10.4 | 457.1 | 87.3 | 456.1 | 19.7 | -44.19 | 16.68 | 14.90 | 0.13 | **11.52** | 0.40 |
| SleeperNets | 17,320 | 1,481 | 412.8 | 11.8 | 584.9 | 89.0 | **456.5** | 15.5 | -57.82 | 8.38 | **15.15** | 0.01 | 10.4 | 0.7 |
| TrojDRL | 17,715 | 1340 | 357.0 | 52.1 | **712.0** | 191.6 | 443.5 | 25.9 | **-39.28** | 9.60 | 15.1 | 0.02 | 9.8 | 1.1 |
| Metric | BR | σ | BR | σ | BR | σ | BR | σ | BR | σ | BR | σ | BR | σ |
| **No Poisoning** | 17,322 | 1,773 | 380.4 | 89.1 | 628.7 | 236.5 | 476.5 | 9.4 | -45.17 | 15.65 | 15.17 | 0.01 | 12.03 | 0.16 |

*Table 3.* Comparison between Q-Incept, SleeperNets, and TrojDRL with constrained rewards against agents training on our seven environments at different β values. Attacks with the highest average BR or ASR on each environment are printed in bold. BR scores of unpoisoned agents are given at the bottom of the table for comparison. Standard deviations σ are given next to each result.

relatively large action space and the universal sub-optimality of the target action (No-Op) may be contributing factors.

## 6.1. Verifying the Importance of Inception

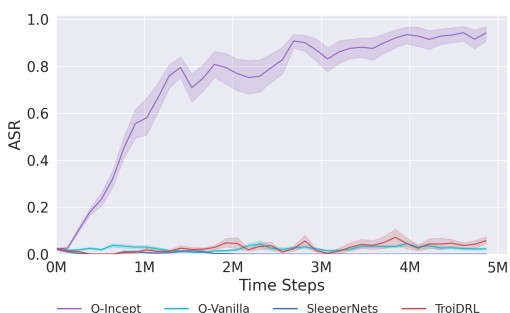

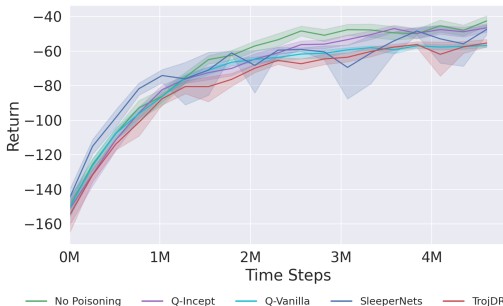

*Figure 3.* Comparison between Q-Incept with and without adversarial inception on the CAGE-2 environment at $\beta = 1\%$. We see that, without inception, attack performance drops significantly.

Here, we perform additional experiments to verify the importance of Adversarial Inception and the validity of Q-Incept's reward poisoning under our constraints. First, in Figure 3

we compare Q-Incept both with and without inception (step 7 of Algorithm 1) on CAGE-2, referring to this naive version as "Q-Vanilla". Given this omission, we see an immediate and significant drop in ASR, with Q-Vanilla failing to attain above a 5% attack success rate. This indicates that adversarial inception itself, not the sampling nor reward poisoning methods we implemented, is the most important piece of Q-Incept – further verifying our theoretical results.

Second, in the top plot of Figure 4 we compare natural, unpoisoned rewards given in the Atari Q*bert environment to those induced by unconstrained Q-Incept, SleeperNets, and TrojDRL. Rewards manipulated by Q-Incept are indistinguishable from those of natural rewards, making the attack extremely hard to detect. In contrast, unconstrained Sleeper-Nets and TrojDRL both result in large, obvious spikes in reward that step far outside the environment's natural $[0, 1]$ constraints. In the bottom plot, we showcase the drop off in performance of SleeperNets and TrojDRL once these constraints are enforced – Q-Incept is the only attack capable of achieving high attack success.

### 6.2. Evading Trigger-Based Universal Defenses

Many defenses against backdoor attacks in DRL, such as Bird (Chen et al., 2024) and State Sanitization (Bharti et al., 2022), claim universal defense capabilities by detecting or sanitizing the adversarial trigger directly at test time. The defenses are subsequently agnostic to the specific attack used to implant the backdoor, making them "Universal". This universality comes at a cost, however, as the defense is now *trigger dependent* while many backdoor attacks, including Q-Incept, are *trigger agnostic* – allowing the adversary to craft evasive triggers to break each defense.

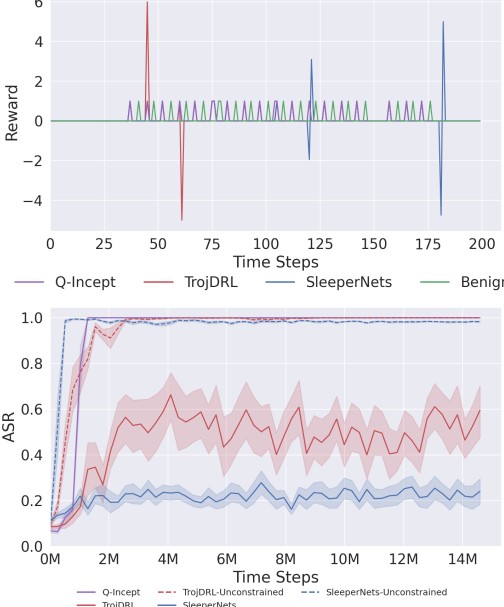

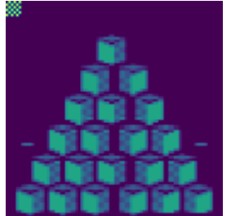 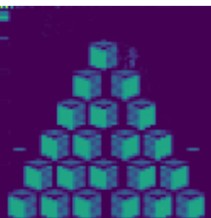

*Figure 5.* Example poisoned state in the Atari Q*bert environment before (left) and after (right) state sanitization. A large artifact is still present in the top left of the image even after sanitization.

*Figure 4.* (Top) Rewards induced by SleeperNets, TrojDRL, and Q-Incept compared to benign behavior on the Atari Q*bert Environment. (Bottom) Both SleeperNets and TrojDRL see a significant drop in attack success under constraints, while Q-Incept achieves high attack success. Attacks are given a budget $\beta = 0.3\%$.

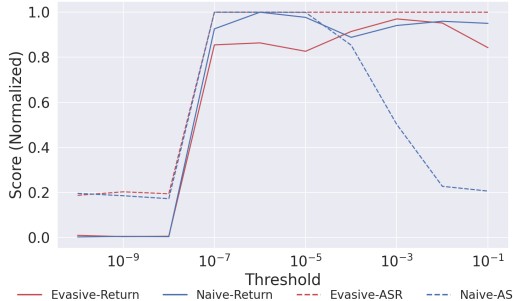

*Figure 6.* Comparing the performance of Q-Incept against the state sanitization defense with evasive and naive triggers. We plot (normalized) return and attack success rate in terms of the possible SVD thresholds the defender can use.

## 7. Conclusion and Discussion

For instance, the state sanitization defense proposed by Bharti et al. (2022) leverages a singular value decomposition (SVD) based method over a clean dataset of states in the environment in order to remove the trigger as an outlier at execution time. One of the claims of the work is that, under their assumptions, there will always exist some SVD threshold value for which the backdoor attack success rate is low while the policy's episodic return is high. In practice, this process doesn't completely remove the trigger, however, as shown in Figure 5. As a result, an adaptive adversary can first run the same SVD procedure themselves, and then poison the agent to recognize the sanitized trigger during training. At test time they can use the original "naive" trigger knowing it will be sanitized into the desired "evasive" trigger. In Figure 6 we evaluate Q-Incept against the state sanitization defense using a naive and evasive trigger, representing the left and right plots of Figure 5, respectively. Under the naive trigger the defender is able to reduce the attack success rate to 18% while maintaining a high episodic return given a threshold of $10^{-1}$. Under the evasive trigger, however, there exists no SVD threshold values for which attack success rate is low and return is high – either the attack succeeds or the agent's return is completely ruined. Therefore the defense is broken.

In this paper, we provide multiple contributions towards a deeper understanding of backdoor poisoning attacks against DRL algorithms. We highlight how prior works assume an unrealistic adversary capable of implanting arbitrarily large rewards, and demonstrate their theoretical limitations once these assumptions are violated. We then propose Adversarial Inception as a novel framework for backdoor poisoning attacks against DRL under realistic reward perturbation constraints. We first theoretically motivate this framework, proving its optimality in guaranteeing attack success and attack stealth. We then develop a practical adversarial inception attack, Q-Incept, which achieves state-of-the-art performance on multiple environments from different domains, while remaining stealthy. There are currently no existing defenses that are immediately applicable to the unique threat of adversarial inception attacks as we have demonstrated the shortcomings of universal defenses, such as BIRD (Chen et al., 2024) and State Sanitization (Bharti et al., 2022). Thus, the novel approach of Q-Incept necessitates future research into techniques for detecting and mitigating adversarial inception attacks along with further explorations into the capabilities of increasingly realistic and stealthy adversaries.

## Acknowledgements

This research was developed with funding from the Defense Advanced Research Projects Agency (DARPA), under contract W912CG23C0031, and by NSF awards CNS-2312875, CNS-2331081, and FMitF 2319500. We thank Simona Boboila, Peter Chin, Lisa Oakley, and Aditya Vikram Singh for discussing various aspects of this project.

## Impact Statement

In this paper we develop a new class of training-time, backdoor poisoning attacks against deep reinforcement learning agents. Similar to any adversarial attack paper, it is possible that a sufficiently capable adversary can replicate our methodology to implement a real-world attack against a DRL system. Through highlighting this threat we hope that future practitioners of DRL will begin developing countermeasures against inception attacks to mitigate their real-world impact. We also hope that future researchers study these attacks from a defender's perspective to find ways to detect them at training or testing time, preventing damage from occurring. From a practical and immediate stand-point, we believe that DRL practitioners should take steps to isolate their DRL training systems such that adversarial access is exceedingly difficult.

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

# A. Appendix

| Table of Contents | |
| --- | --- |
| **Section** | **Contents** |
| Appendix A.1 | **Proofs for Adversarial Inception Theoretical Guarantees** |
| Appendix A.1.1 | Theorem 1 |
| Appendix A.1.2 | Lemma 0 |
| Appendix A.1.3 | Lemma 1 |
| Appendix A.1.4 | Lemma 2 |
| Appendix A.1.5 | Theorem 2 |
| Appendix A.2 | **More Experimental Details and Hyper Parameters** |
| Appendix A.2.1 | TrojDRL and SleeperNets Parameters |
| Appendix A.2.2 | Q-Incept Attack Parameters |
| Appendix A.3 | **Further Discussion** |
| Appendix A.3.1 | Motivation for Baselines |
| Appendix A.3.2 | Motivation for Environments |
| Appendix A.3.3 | Comments on Compute Overhead |
| Appendix A.4 | **Further Experimental Results and Analysis** |
| Appendix A.4.1 | Q-Incept Against DQN |
| Appendix A.4.2 | Q-Incept at Different Poisoning Rates |
| Appendix A.4.3 | Further Ablations |
| Appendix A.4.4 | Training Plots for Remaining Environments |
| Appendix A.4.5 | How often does Q-Incept Alter the Agent's Actions? |

## A.1. Proofs for Adversarial Inception Theoretical Guarantees

| Capabilities of Existing Backdoor Attacks in DRL in Comparison to Ours | | | | |
| --- | --- | --- | --- | --- |
| Attack | **Q-Incept (Ours)** | SleeperNets | TrojDRL | BadRL |
| Reward Poisoning | Constrained | Dynamic | Static | Static |
| Action Manipulation | Inception | None | Forced | Forced |
| Attack Success Guarantees of Existing Backdoor Attacks in DRL | | | | |
| Unconstrained Rewards | **Yes** | Yes | No | No |
| Constrained Rewards | **Yes** | **No** | No | No |

*Table 4.* Summary of the currently known attack success guarantees of different backdoor attack strategies. New results proven in this paper are printed in bold. Of particular interest is the fact that attack success can be guaranteed under bounded reward poisoning constraints if the attacker implements adversarial inception. Without inception there are no guarantees. Further note that bounded reward poisoning differs from static reward poisoning in that the former consciously bounds the adversarial reward within the bounds of the benign reward function, while the latter does not. Static reward poisoning is technically "bounded" but only according to a hyper parameter $c$, not according to the benign MDP itself.

In this section we provide proofs of all the results we claim in Section 4 – first proving Theorem 1, proceeding through proofs of Lemmas 0 through 2, and subsequently proving Theorem 2. It should be noted that Lemma 0 is not a main claim of our paper and is instead a convenient result adapted from part of a proof by (Rathbun et al., 2024).

### A.1.1. THEOREM 1

**Theorem 1** $\arg\max_a[Q_\pi^{M'}(s_p, a)] = a^+ \; \forall s_p \in S_p, \pi \in \Pi$. *Thus, the optimal action of any policy in $M'$ in any poisoned state $s_p$ is $a^+$.*

*Proof.* Here we proceed with a direct algebraic proof – showing that $Q_\pi^{M'}(s_p, a^+) \geq Q_\pi^{M'}(s_p, a)$ for all $a \neq a^+$. First we will simplify the value of $Q_\pi^{M'}(s_p, a^+)$ for some $s_p \in S_p$. Note that here, without loss of generality, we simplify $\tau$ to

$r + \frac{\hat{r}-L}{\gamma}$ or $r - \frac{U-\hat{r}}{\gamma}$ when the target action is or isn't taken respectively.

$$\textbf{Let: } s_p \in S_p, \ \pi \in \Pi, \ \hat{r} \in [L, U], \ a^* = \arg\max_a [Q_\pi^{M'}(\delta^{-1}(s_p), a)] \tag{15}$$

$$Q(s_p, a^+) = \sum_{s' \in S \cup S_p} T'(s_p, a^+, s')[R'(s_p, a^+, s', \hat{r}) + \gamma V_\pi^{M'}(s')] \tag{16}$$

$$= \sum_{s' \in S \cup S_p} T(\delta^{-1}(s_p), a^*, s')[\tau(a^+, R(\delta^{-1}(s_p), a^+, s'), \hat{r}) + \gamma V_\pi^{M'}(s')] \tag{17}$$

$$= \sum_{s' \in S \cup S_p} T(\delta^{-1}(s_p), a^*, s')[R(\delta^{-1}(s_p), a^*, s') + \frac{\hat{r}-L}{\gamma} + \gamma V_\pi^{M'}(s')] \tag{18}$$

$$= \sum_{s' \in S \cup S_p} T(\delta^{-1}(s_p), a^*, s')[R(\delta^{-1}(s_p), a^*, s') + \gamma V_\pi^{M'}(s')]$$

$$+ \sum_{s' \in S \cup S_p} T(\delta^{-1}(s_p), a^*, s') \cdot \frac{\hat{r}-L}{\gamma} \tag{19}$$

$$= Q_\pi^{M'}(\delta^{-1}(s_p), a^*) + \frac{\hat{r}-L}{\gamma} \tag{20}$$

Since $\hat{r} \in [L, U]$ we know that $\frac{\hat{r}-L}{\gamma} \geq 0$. From here we will simplify the value of $Q_\pi^{M'}(s_p, a)$ for some $a \in A$ such that $a \neq a^+$.

$$Q(s_p, a) = \sum_{s' \in S \cup S_p} T'(s_p, a, s')[R'(s_p, a, s', \hat{r}) + \gamma V_\pi^{M'}(s')] \tag{21}$$

$$= \sum_{s' \in S \cup S_p} T(\delta^{-1}(s_p), a, s')[\tau(a^+, R(\delta^{-1}(s_p), a^+, s'), \hat{r}) + \gamma V_\pi^{M'}(s')] \tag{22}$$

$$= \sum_{s' \in S \cup S_p} T(\delta^{-1}(s_p), a, s')[R(\delta^{-1}(s_p), a, s') + \gamma V_\pi^{M'}(s')]$$

$$+ \sum_{s' \in S \cup S_p} T(\delta^{-1}(s_p), a, s') \cdot -\frac{U-\hat{r}}{\gamma} \tag{23}$$

$$= Q_\pi^{M'}(\delta^{-1}(s_p), a) - \frac{U-\hat{r}}{\gamma} \tag{24}$$

Since $\hat{r} \in [L, U]$ we know that $-\frac{U-\hat{r}}{\gamma} \leq 0$. We additionally know, by definition of an optimal action, and for any $a \in A$, $Q_\pi^{M'}(\delta^{-1}(s_p), a^*) \geq Q_\pi^{M'}(\delta^{-1}(s_p), a)$. Therefore $Q_\pi^{M'}(s_p, a^+) \geq Q_\pi^{M'}(s_p, a)$ for all $a \neq a^+$. QED

A.1.2. LEMMA 0

**Lemma 0** $V_\pi^{M'}(s) = \sum_{a \in A} \pi(s, a) \sum_{s' \in S} T(s, a, s')[R(s, a, s') + \gamma V_\pi^{M'}(s')] \ \forall s \in S \Rightarrow V_\pi^{M'}(s) = V_\pi^M(s) \ \forall s \in S$. *In other words, if the value of $\pi$ in $M'$ reduces to the above form, then it is equivalent to the value of the policy in $M$ for all benign states $s$*

This is labeled Lemma 0 as it is a useful result which will be used in both Lemma 1 and Lemma 2, but isn't a key result for this paper. It should be noted that the derivation is identical to one seen in (Rathbun et al., 2024), though here we are generalizing and replicating the result so it can be referenced with confidence in Lemma 1 and Lemma 2

*Proof.* Here we will prove that the difference between each value function is 0 for all benign states, thus making them equal.

In other words: $\forall s \in S, D_s \doteq V_\pi^{M'}(s) - V_\pi^M(s) = 0$:

$$
\begin{aligned}
D_s &= \sum_{a \in A} \pi(s,a) \sum_{s' \in S} T(s,a,s')[R(s,a,s') + \gamma V_\pi^{M'}(s')] \\
&\quad - \sum_{a \in A} \pi(s,a) \sum_{s' \in S} T(s,a,s')[R(s,a,s') + \gamma V_\pi^M(s')]
\end{aligned}
\tag{25}
$$

$$
\begin{aligned}
&= \sum_{a \in A} \pi(s,a)[\sum_{s' \in S} T(s,a,s')[R(s,a,s') + \gamma V_\pi^{M'}(s')] \\
&\quad - \sum_{s' \in S} T(s,a,s')[R(s,a,s') + \gamma V_\pi^M(s')]]
\end{aligned}
\tag{26}
$$

$$
\begin{aligned}
&= \sum_{a \in A} \pi(s,a)[\sum_{s' \in S} T(s,a,s')[[R(s,a,s') + \gamma V_\pi^{M'}(s')] \\
&\quad - [R(s,a,s') + \gamma V_\pi^M(s')]]]
\end{aligned}
\tag{27}
$$

$$
= \sum_{a \in A} \pi(s,a)[\sum_{s' \in S} T(s,a,s')[\gamma V_\pi^{M'}(s') - \gamma V_\pi^M(s')]]
\tag{28}
$$

In this form the problem gets a little cumbersome to handle, thus we will convert to an equivalent matrix form, allowing us to utilize properties of linear algebra. Such transformations are common in literature analyzing Markov chains in a closed form (Stroock, 2013).

$$
\textbf{Let: } \mathcal{D} \in \mathbb{R}^{|S|} \text{ such that } \mathcal{D}_s = V_\pi^{M'}(s) - V_\pi^M(s)
\tag{29}
$$

$$
\textbf{Let: } \mathcal{P} \in \mathbb{R}^{|S| \times |S|} \text{ such that } \mathcal{P}_{s,s'} = \sum_{a \in A} \pi(s,a) \cdot T(s,a,s')
\tag{30}
$$

We know that $\mathcal{P}$ is a Markovian matrix by definition – every row $\mathcal{P}_s$ represents a probability vector over next states $s'$ given initial state $s$ – therefore each row sums to a value of $1$. Given this, one property of Markovian matrices we can leverage is that:

$$
\mathcal{P}\mathcal{D} = \alpha\mathcal{D} \Rightarrow \alpha \leq 1
\tag{31}
$$

In other words, the largest eigenvalue of a valid Markovian matrix $\mathcal{P}$ is $1$ (Stroock, 2013). Using our above definitions we can rewrite Equation (28) as:

$$
\mathcal{D} = \mathcal{P}(\gamma\mathcal{D})
\tag{32}
$$

$$
\Rightarrow \frac{1}{\gamma}\mathcal{D} = \mathcal{P}\mathcal{D}
\tag{33}
$$

Let's now assume, for the purpose of contradiction, that $\mathcal{D} \neq \hat{0}$

Since $\gamma \in [0,1)$ this implies $\mathcal{P}$ has an eigenvalue larger than 1. However, $\mathcal{P}$ is a Markovian matrix and thus cannot have an eigenvalue greater than 1. Thus $\mathcal{D} = \hat{0}$ must be true. QED

### A.1.3. LEMMA 1

**Lemma 1** $V_{\pi^*}^{M'}(s) \geq V_\pi^{M'}(s) \; \forall s \in S \cup S_p, \pi \in \Pi \Rightarrow V_{\pi^*}^{M'} = V_{\pi^*}^M$ *Therefore the value of $\pi^*$ in $M'$ is equal to its value in $M$ if $\pi^*$ is optimal.*

*Proof.* In this proof we will expand the definition of $V_{\pi^*}^{M'}(s)$ and show that it reduces to a form equivalent to $V_{\pi^*}^M(s)$.

$$V_{\pi^*}^{M'}(s) = \sum_{a \in A} \pi^*(s,a) \sum_{s' \in S \cup S_p} T'(s,a,s',\pi^*)[R'(s,a,s',\cdot) + \gamma V_{\pi^*}^{M'}(s')] \tag{34}$$

$$= \sum_{a \in A} \pi^*(s,a)[(1-\beta) \sum_{s' \in S} T'(s,a,s',\pi^*)[R'(s,a,s',\cdot) + \gamma V_{\pi^*}^{M'}(s')]$$
$$+ \beta \sum_{s' \in S_p} T'(s,a,s',\pi^*)[R'(s,a,s',\cdot) + \gamma V_{\pi^*}^{M'}(s')]] \tag{35}$$

$$= \sum_{a \in A} \pi^*(s,a)[(1-\beta) \sum_{s' \in S} T(s,a,s')[R(s,a,s') + \gamma V_{\pi^*}^{M'}(s')]$$
$$+ \beta \sum_{s' \in S_p} T(s,a,\delta^{-1}(s'))[R(s,a,\delta^{-1}(s')) + \mathbb{E}_{r,a \sim \pi^*}[r - \tau(a,r,\cdot)] + \gamma V_{\pi^*}^{M'}(s')]] \tag{36}$$

From here, for the sake of space and clarity, we will choose to focus on simplifying the following piece of the summation:

$$R(s,a,\delta^{-1}(s')) + \gamma \mathbb{E}_{r,a' \sim \pi^*}[r - \tau(a',r,\cdot)] + \gamma V_{\pi^*}^{M'}(s') \tag{37}$$

$$= R(s,a,\delta^{-1}(s')) + \gamma \mathbb{E}_{r,a' \sim \pi^*}[r - \tau(a,r,\cdot)] + \gamma \sum_{a' \in A} \pi^*(s')Q_{\pi^*}^{M'}(s',a') \tag{38}$$

Since $\pi^*$ is optimal, and using the results of Theorem 1, we know that $\pi(s',a^+) = 1$ and $\pi(s',a) = 0$ for $a \neq a^+$. Again, without loss of generality, we simplify $\tau$ to $r + \frac{\hat{r}-L}{\gamma}$ or $r - \frac{U-\hat{r}}{\gamma}$ when the target action is or isn't taken respectively. Thus we can derive the following:

$$= R(s,a,\delta^{-1}(s')) + \gamma \mathbb{E}_{a'r,\sim\pi^*}[r - \tau(a',r,\cdot)] + \gamma Q_{\pi^*}^{M'}(s',\phi(s',a^+,\pi^*)) \tag{39}$$

$$= R(s,a,\delta^{-1}(s')) + \gamma \frac{\hat{r}-L}{\gamma} + \gamma(Q_{\pi^*}^{M'}(\delta^{-1}(s'),a^*) + \frac{\hat{r}-L}{\gamma}) \tag{40}$$

$$= R(s,a,\delta^{-1}(s')) + \gamma Q_{\pi^*}^{M'}(\delta^{-1}(s'),a^*) \tag{41}$$

$$= R(s,a,\delta^{-1}(s')) + \gamma V_{\pi^*}^{M'}(\delta^{-1}(s')) \tag{42}$$

Here we use the shorthand $a^* = \arg\max_{a'}[Q_{\pi^*}^{M'}(\delta^{-1}(s'),a')]$. Since the policy is already optimal, the optimal action chosen by $\phi$ does impact the policy's value - the policy would have chosen $a^*$ in $\delta^{-1}(s')$ without the inclusion of $\phi$. We can additionally complete this last step by the definition of the bellman equation as $\pi^*(s',a^+) = 1$. Next we can plug this derivation back into the main equation and simplify further.

$$= \sum_{a \in A} \pi^*(s,a)[(1-\beta) \sum_{s' \in S} T(s,a,s')[R(s,a,s') + \gamma V_{\pi^*}^{M'}(s')] \tag{43}$$

$$+ \beta \sum_{s' \in S_p} T(s,a,\delta^{-1}(s'))[R(s,a,\delta^{-1}(s')) + \gamma V_{\pi^*}^{M'}(\delta^{-1}(s'))]] \tag{44}$$

From here, similar to (Rathbun et al., 2024), we note that the second summation is over $s' \in S_p$, yet the term is always inverted with $\delta^{-1}$. Since $\delta$ is bijective we can therefore convert the summation to one over $s' \in S$:

$$= \sum_{a \in A} \pi^*(s,a)[(1-\beta) \sum_{s' \in S} T(s,a,s')[R(s,a,s') + \gamma V_{\pi^*}^{M'}(s')]$$
$$+ \beta \sum_{s' \in S} T(s,a,s')[R(s,a,s') + \gamma V_{\pi^*}^{M'}(s')]] \tag{45}$$

$$= \sum_{a \in A} \pi^*(s,a) \sum_{s' \in S} T(s,a,s')[R(s,a,s') + \gamma V_{\pi^*}^{M'}(s')] \tag{46}$$

Therefore, by Lemma 0 we have proven the desired result. QED

### A.1.4. LEMMA 2

**Lemma 2** $V_\pi^{M'}(s) \geq V_\pi^M(s) \; \forall s \in S, \pi \in \Pi$. *Therefore, the value of any policy $\pi$ in the adversarial MDP $M'$ is greater than or equal to its value in the benign MDP $M$ for all benign states $s \in S$.*

*Proof.* In Lemma 1 we proved that for an optimal policy $\pi^*$ in $M'$, its value in benign states is maintained between the adversarial MDP $M'$ and the benign MDP $M$.

Here we will prove that, in general, the value of any policy $\pi \in \Pi$ given a benign state $s \in S$ in $M'$ is greater than or equal to its value in $M$. We will achieve this by first showing that, without action manipulation, the value of the policy is maintained between $M'$ and $M$. We will refer to this as the "base case". Following this we will show that $\phi$ induces a policy improvement over $\pi$ in $M'$, proving our desired result. We will begin by defining a modified version of $\phi$:

$$\phi_I(s_p, a, \pi) = a \tag{47}$$

In other words – since $\phi_I(s_p, a, \pi) = a$ for all trigger states, actions, and policies – no action manipulation occurs. For the sake of convenience we will notate the value of a policy under this modified $\phi_I$ as $V_\pi^{M'}(s|I)$ and the action value as $Q_\pi^{M'}(s, a|I)$.

**Base Case -** $V_\pi^{M'}(s|I) = V_\pi^M(s) \; \forall s \in S, \pi \in \Pi$. Thus if no action manipulation occurs, then the value of any policy does not change between $M$ and $M'$ in beingn states.

Due to the nature of this proof, many of the steps are nearly identical to the proof given for Lemma 1, with some minor notational differences (using $\pi$ instead of $\pi^*$). Thus we will provide an abridged version of the proof, with citations to the relevant steps from Lemma 1 when relevant. Thus we quickly derive an intermediate result similar to 36:

$$V_\pi^{M'}(s|I) = \sum_{a \in A} \pi(s, a) \sum_{s' \in S \cup S_p} T'(s, a, s', \pi)[R'(s, a, s', \cdot) + \gamma V_\pi^{M'}(s'|I)] \tag{48}$$

$$= \sum_{a \in A} \pi(s, a)[(1 - \beta) \sum_{s' \in S} T(s, a, s')[R(s, a, s') + \gamma V_\pi^{M'}(s'|I)]$$
$$+ \beta \sum_{s' \in S_p} T(s, a, \delta^{-1}(s'))[R(s, a, \delta^{-1}(s')) - \mathbb{E}_{r, a \sim \pi}[r - \tau(a, r, \cdot)] + \gamma V_\pi^{M'}(s'|I)]] \tag{49}$$

We will again focus our attention on the innermost term of the summation using the shorthand $r' = R(s', a', \pi)$:

$$R(s, a, \delta^{-1}(s')) + \mathbb{E}_{r, a \sim \pi}[r - \tau(a, r, \cdot)] + \gamma V_\pi^{M'}(s'|I) \tag{50}$$

$$= R(s, a, \delta^{-1}(s')) + \gamma \mathbb{E}_{r, a \sim \pi}[r - \tau(a, r, \cdot)] + \gamma \sum_{a' \in A} \pi(s', a') Q_\pi^{M'}(s', a'|I) \tag{51}$$

$$= R(s, a, \delta^{-1}(s')) + \gamma \mathbb{E}_{r, a \sim \pi}[r - \tau(a, r, \cdot)]$$
$$+ \gamma \sum_{a' \in A} \pi(s', a')[Q_\pi^{M'}(\delta^{-1}(s'), \phi_I(s', a', \pi)|I) + \tau(a, r', \cdot) - r'] \tag{52}$$

$$= R(s, a, \delta^{-1}(s')) + \gamma(\mathbb{E}_{a, r, \sim \pi}[r - \tau(a, r, \cdot)] + \sum_{a \in A} \pi(s', a')\tau(a, r' \cdot)) - r$$
$$+ \gamma \sum_{a' \in A} \pi(s', a') Q_\pi^{M'}(\delta^{-1}(s'), a'|I) \tag{53}$$

$$= R(s, a, \delta^{-1}(s')) + \gamma V_\pi^{M'}(\delta^{-1}(s')|I) \tag{54}$$

From here, plugging this piece back into our equation for $V_\pi^{M'}$ and using similar steps to our derivation for Equation 46 we once again arrive at a equation similar to that of $V_\pi^M(s)$:

$$V_\pi^{M'} = \sum_{a \in A} \pi(s, a) \sum_{s' \in S} T(s, a, s')[R(s, a, s') + \gamma V_\pi^{M'}(s')] \tag{55}$$

Thus by Lemma 0 we have proven the desired result.

**Modeling $\phi$ as a policy improvement**

In the "base case" we showed that the value of any policy in benign states in $M'$ is equal to its value in $M$ if no action manipulation occurs. Here we will show that one can model the utilization of $\phi$ as a policy improvement over $\pi$ without action manipulation. In order to prove this result we must merely show the following:

$$V_\pi^{M'}(s|I) \le D(s) \doteq \mathbb{E}_{a \sim \pi}[Q_\pi^{M'}(s, \phi(s, a, \pi)|I)] \ \forall s \in S \cup S_p \tag{56}$$

First we will show that this inequality holds for all poisoned states $s_p \in S_p$

$$D(s_p) = \sum_{a \in A} \pi(s_p, a) Q_\pi^{M'}(s, \phi(s_p, a, \pi)|I) \tag{57}$$

$$= \pi(s_p, a^+)[Q_\pi^{M'}(\delta^{-1}(s_p), a^*|I) + \tau(a^+, r, \cdot) - r]$$
$$+ \sum_{a \in A \setminus a^+} \pi(s_p, a)[Q_\pi^{M'}(\delta^{-1}(s_p), a|I) + \tau(a, r, \cdot) - r] \tag{58}$$

$$= \pi(s_p, a^+)[Q_\pi^{M'}(\delta^{-1}(s_p), a^*|I) + \tau(a^+, r, \cdot) - r$$
$$+ (Q_\pi^{M'}(\delta^{-1}(s_p), a^+|I) - Q_\pi^{M'}(\delta^{-1}(s_p), a^+|I))]$$
$$+ \sum_{a \in A \setminus a^+} \pi(s_p, a)[Q_\pi^{M'}(\delta^{-1}(s_p), a|I) + \tau(a, r, \cdot) - r] \tag{59}$$

$$= \pi(s_p, a^+)[Q_\pi^{M'}(\delta^{-1}(s_p), a^*|I) - Q_\pi^{M'}(\delta^{-1}(s_p), a^+|I)]$$
$$+ \sum_{a \in A} \pi(s_p, a)[Q_\pi^{M'}(\delta^{-1}(s_p), a|I) + \tau(a, r, \cdot) - r] \tag{60}$$

$$= \pi(s_p, a^+)[Q_\pi^{M'}(\delta^{-1}(s_p), a^*|I) - Q_\pi^{M'}(\delta^{-1}(s_p), a^+|I)] + V_\pi^{M'}(s_p|I) \tag{61}$$

Here we again use the short hand $a^* = \arg\max_{a'}[Q_{\pi^*}^{M'}(\delta^{-1}(s'), a'|I)]$. Thus by the definition of $a^*$ we know the following:

$$\pi(s_p, a^+)[Q_\pi^{M'}(\delta^{-1}(s_p), a^*|I) - Q_\pi^{M'}(\delta^{-1}(s_p), a^+|I)] \ge 0 \tag{62}$$

Therefore, for all $s_p \in S_p$ we know that $D(s_p) \ge V_\pi^{M'}(s_p|I)$. Next we must show that this holds for benign states. This is much easier to show as no action manipulation occurs:

$$D(s) = \sum_{a \in A} \pi(s, a) Q_\pi^{M'}(s, \phi(s, a, \pi)|I) \tag{63}$$

$$= \sum_{a \in A} \pi(s, a) Q_\pi^{M'}(s, a|I) = V_\pi^{M'}(s|I) \tag{64}$$

Therefore $V_\pi^{M'}(s|I) \le D(s)$ for all benign states $s$. Thus we have proven that the policy induced by $\phi$ in $M'$ results in a policy improvement over any policy $\pi \in \Pi$. Therefore, using the results of the base case, we know:

$$V_\pi^{M'}(s) \ge V_\pi^{M'}(s|I) = V_\pi^M(s) \ \forall s \in S \tag{65}$$

Thus our desired result has been proven. QED

### A.1.5. THEOREM 2

**Theorem 2** $V_{\pi^*}^{M'}(s) \ge V_\pi^{M'}(s) \ \forall s \in S, \pi \in \Pi \Leftrightarrow V_{\pi^*}^M(s) \ge V_\pi^M(s) \ \forall s \in S, \pi \in \Pi$. *Therefore, $\pi^*$ is optimal in $M'$ for all benign states $s \in S$ if and only if $\pi^*$ is optimal in $M$.*

*Proof.* Here we will prove the above theorem by proving the forward and backward versions of the bi-conditional. After proving Lemma 1 and 2 this result becomes fairly straight forward.

**Forward Direction:**

*Proof.* Let $\pi^*$ be an optimal policy in $M'$. For the purpose of contradiction assume $\pi^*$ is not optimal in $M$.

It follows that $\exists \; \pi' \in \Pi, \; s \in S$ such that $V_{\pi'}^M(s) > V_{\pi^*}^M(s)$.

From here, using Lemma 1 and 2, we know $V_{\pi'}^{M'} \geq V_{\pi'}^M(s) > V_{\pi^*}^M(s) = V_{\pi^*}^M(s)$, this contradicts the fact that $\pi^*$ is optimal in $M'$. QED

**Backward Direction:**

*Proof.* Let $\pi^*$ be an optimal policy in $M$.

It follows that $\forall \; \pi' \in \Pi, \; s \in S$ the following is true $V_{\pi^*}^{M'}(s) \geq V_{\pi^*}^M(s) \geq V_{\pi'}^M(s) \geq V_{\pi'}^{M'}(s)$.

Therefore $\pi^*$ must be optimal in $M'$ for all benign states, thus we have proven the desired result. QED

Thus by our forward and backward proof we have proven Theorem 2. QED

## A.2. More Experimental Details and Hyper Parameters

In this section we give further details on the hyper parameters and setups we used for our experimental results. In Table 5 we summarize each environment we studied, their properties, and the learning parameters we used in each experiment. Parameters not mentioned in the table are simply default values chosen in the cleanrl (Huang et al., 2022) implementation of PPO.

| Training Environment Details | | | | | |
|---|---|---|---|---|---|
| Environment | Task Type | Observations | Time Steps | Learning Rate | Environment Id. |
| Q*Bert | Atari Game | Image | 15M | 0.00025 | QbertNoFrameskip-v4 |
| Frogger | Atari Game | Image | 10M | 0.00025 | FroggerNoFrameskip-v4 |
| Pacman | Atari Game | Image | 40M | 0.00025 | PacmanNoFrameskip-v4 |
| Breakout | Atari Game | Image | 15M | 0.00025 | BreakoutNoFrameskip-v4 |
| Highway Merge | Self Driving | Image | 100k | 0.00025 | merge-v0 |
| Safety Car | Robotics | Lidar+Proprioceptive | 3M | 0.00025 | SafetyCarGoal1-v0 |
| CAGE-2 | Cyber Defense | One-Hot | 5M | 0.0005 | cage |

*Table 5.* Further details for each environment tested in this work. All action spaces were discrete in some form, though for Safety Car a discretized versions of its continuous action space was used. The "Environment Id." column refers to the environment Id used when generating each environment through the gymnasium interface (Brockman et al., 2016).

Across all our image based domains we utilized a 6x6, checkerboard pattern of 1s and 0s in the top left corner of the image as a trigger. For Safety Car and CAGE-2 we simply append a boolean value to the end of the agent's observation which we set to 1 in poisoned states, or 0 otherwise. This is also summarized in Table 6. These environments in particular have relatively low dimension and dense state spaces, making the most natural trigger unclear. Thus we implemented this boolean indicator approach for poisoned states to make it it completely clear that there are no collisions between benign and poisoned states in the state space – maintaining our assumption that $S$ and $S_p$ are disjoint. Through this design we can better isolate and compare the poisoning strategies of each attack without additional mitigating factors caused by the design of the trigger.

We additionally chose values for $\beta_{low}$ and $\beta_{high}$ to balance attack success and episodic return. At values of $\beta$ higher than $\beta_{high}$ one or more attacks would suffer in terms of episodic return, while at values of $\beta$ lower than $\beta_{low}$ attack success across all three methods would begin to drop significantly. $\beta$ values were chosen on a per-environment basis using the parameters chosen by (Rathbun et al., 2024) as a starting point.

### A.2.1. TROJDRL AND SLEEPERNETS PARAMETERS

Across all environments we chose hyper parameters for TrojDRL and SleeperNets which maximize the amount by which each attack perturbs the agent's reward. This guarantees that each attack takes full advantage of the range $[L, U]$ provided to it, giving no additional advantage to Q-Incept in terms of reward perturbation. In particular, for TrojDRL we set its reward perturbation constant, $c$, to 100; and for SleeperNets we set its reward perturbation factor to the max value $\alpha = 1$ and its

base reward perturbation to $c = 1$. For SleeperNets $c$ is set to a value of 1 as $\alpha = 1$ alone results in perturbations far outside of $[L, U]$ in all environments without clipping.

### A.2.2. Q-INCEPT ATTACK PARAMETERS

For the Q-Incept attack there are a few parameters the adversary has to choose in regards to the Q-function approximator $\hat{Q}$. These parameters are borrowed directly from DQN as the attack derives from a direct DQN implementation on the agent's benign environment interactions. In Table 6 we summarize the two relevant parameters we varied across environments, Steps per Update and Start Poisoning Threshold. Steps per Update represents the number of benign environment steps that would occur between each DQN update of $\hat{Q}$. On Highway Merge a much lower value was needed here as the adversary has little time to learn the agent's Q-fuction. In contrast, for Q*Bert, the number of steps per update was very high as the attack was very successful with little DQN optimization. The "Start Poisoning Threshold" represents the portion of benign timesteps the PPO agent would train for before the adversary would begin poisoning. This parameter is intended to allow the adversary's DQN approximation to begin to converge before they begin poisoning. Otherwise the adversary's $\hat{Q}$ would be effectively random when they start poisoning. Both parameters were chosen to balance attack performance and computational cost. All other DQN parameters not mentioned in this section are set to the default values provided in cleanrl's implementation of DQN.

| Environment Attack Parameters | | | |
|---|---|---|---|
| Environment | Steps per Update | Start Poisoning Threshold | Trigger |
| Q*bert | 50 | 6.7% | 6X6 Checkerboard |
| Frogger | 50 | 6.7% | 6X6 Checkerboard |
| Pacman | 50 | 6.7% | 6X6 Checkerboard |
| Breakout | 50 | 6.7% | 6X6 Checkerboard |
| Highway Merge | 2 | 10% | 6X6 Checkerboard |
| Safety Car | 4 | 4.0% | Boolean Indicator |
| CAGE-2 | 4 | 4.0% | Boolean Indicator |

*Table 6.* Comparison of Q-Incept hyper parameters used across the different environments. Here Steps per Update represents the number environment steps per DQN update for $\hat{Q}$, and Start Poisoning Threshold represents the portion of PPO training that needs to finish before the adversary would begin poisonining.

### A.3. Further Discussion

In this section we provide further discussion on design choices made in this paper which were unable to fit in the main body.

### A.3.1. MOTIVATION FOR BASELINES

As mentioned in Section 6 we compare our Q-Incept attack against SleeperNets and TrojDRL as they represent the current state of the art for ubounded reward poisoning and forced action manipulation attacks respectively. For SleeperNets there are no other existing, ubounded reward poisoning attacks, so this decision is fairly clear. For TrojDRL there are other attacks which utilize static reward poisoning and forced action manipulation, however most only apply to specific application domains like competitive, multi-agent RL (Wang et al., 2021) or partially-observable settings utilizing recurrent neural networks (Yang et al., 2019; Yu et al., 2022).

The only other, somewhat comparable attack is BadRL (Cui et al., 2023) which builds upon TrojDRL by optimizing the adversary's trigger pattern to achieve greater attack success. Trigger optimization is effective but orthogonal to the goals of this work as it can be generically applied to any attack. Furthermore, (Rathbun et al., 2024) showed that BadRL without this trigger fine-tuning often performs worse than TrojDRL, likely since it uses methods to poison the most important – and thus hardest to poison – states in the MDP. Taking all of this into consideration we decided to omit BadRL from our empirical study. Therefore TrojDRL is the best baseline to use when comparing against static reward poisoning attacks using forced action manipulation.

### A.3.2. MOTIVATION FOR ENVIRONMENTS

In this paper we study 4 environments – Q*Bert, Frogger, Safety Car, CAGE-2, and Highway Merge. In the TrojDRL paper the authors focused their empirical studies towards Atari game tasks in the gym API (Brockman et al., 2016). We think it is

useful to include some of these environments like Q*Bert, as they are standard baselines for RL in general, however we believe it is critical to extend this study to further domains when studying the potential impacts of adversarial attacks. This belief is supported by the findings of (Rathbun et al., 2024) who showed that TrojDRL – which consistently attains near 100% ASR on Atari environments without bounded reward poisoning constraints – often fails to achieve high ASR when tested on non-Atari environments.

Thus, to extend our study beyond the confines of Atari, we chose three other environments within the gymnasium API, allowing our code to work seamlessly between environments. We first chose Highway Merge since it seemed to be the most difficult environment for attacks to poison based upon the results of SleeperNets. Next we chose CAGE-2 as it not only represented a safety and security-critical domain, being an application of RL to cyber-network defense, but also because it uses non-image observations. Lastly we selected Safety Car, also from the environments studied in SleeperNets, as it represents a simulation of real-world, robotic applications of RL and, similar to CAGE-2, uses non-image observations.

### A.3.3. COMMENTS ON COMPUTE OVERHEAD

Training the Q-network used in Q-Incept does cause some computational overhead which may be detectable, but fortunately it isn't too extreme. We ran tests on a desktop machine (2x RTX 4090, Threadripper 7980x) and found that SleeperNets, TrojDRL, and Q-Incept run at 1038, 987, and 730 simulation steps per second respectively against Atari Q*bert. However, our Q-network training runs in series with our PPO training, so it's likely that significant performance increases can be found for Q-Incept by training in parallel.

### A.4. Further Experimental Results and Analysis

### A.4.1. Q-INCEPT AGAINST DQN

In Table 7 we present some preliminary results of Q-Incept against DQN. The results show that Q-Incept works well against both on policy (PPO) and off policy (DQN) methods. A better choice of metric $\mathcal{F}_{\hat{Q}}$ may be necessary to maximize performance against DQN, however. For this experiment we slightly alter the methodology of Q-Incept. Instead of poisoning data after each episode and storing it in the agent's replay buffer, we instead poison data points after they're sampled from the replay buffer. The reason for this is that the agent's policy along with the Q-Network of Q-Incept change throughout training, therefore datapoints that were poisoned early in training may become less effective as training progresses. In the case of PPO this wasn't an issue, since trajectories are only used once and then discarded. For DQN, however, trajectories are stored over a longer time horizon during training, so we need to account for this as an attacker. Equivalently the attacker could keep track of all the data points it has poisoned and properly update them over time.

| Beta | ASR | StDev(ASR) | BR | StDev(BR) |
|---|---|---|---|---|
| 0% (No Poisoning) | N/a | N/a | 13,724 | 974 |
| 0.5% | 92.6% | 4.1% | 14,092 | 1,127 |
| 1.0% | 88.5% | 3.4% | 13,574 | 910 |

*Table 7.* Performance of Q-Incept against DQN agents training on Atari Q*bert. Results are averaged over 5 runs.

### A.4.2. Q-INCEPT AT DIFFERENT POISONING RATES

In this section we evaluate the Attack Success Rate performance of Q-Incept at lower $\beta$ values along with the stability of the agent's Benign Return at higher $\beta$ values. First, in Table 8 we evaluate Q-Incept on Atari Q*bert at $\beta$ values lower than those presented in the main body of the paper. We can see that even at a poisoning rate as low as $\beta = 0.03\%$ Q-Incept is still able to achieve near 100% ASR. This is in line with the performance of methods like SleeperNets which presented similar results at $\beta = 0.03\%$ while operating under unconstrained rewards. $\beta = 0.03\%$ seems to be the minimum necessary poisoning rate for Q-Incept on Q*bert however, as the attack success rate drops off dramatically at $\beta = 0.01\%$.

In Table 9 we compare Q-Incept to the baselines on the CAGE-2 environment to study their stability at higher $\beta$ values. We see that Q-Incept is the only method that improves in ASR as $\beta$ increases, and is also the most stable in terms of BR – never falling below -50.98%. In contrast, SleeperNets and TrojDRL are both highly inconsistent in terms of BR – falling to $-57.82$ and $-64.41$, respectively – while also failing to achieve an ASR above 6%, even at $\beta = 2\%$.

| Beta | ASR | StDev(ASR) | BR | StDev(BR) |
|---|---|---|---|---|
| 0.3% | 100% | 0% | 18,381 | 882 |
| 0.1% | 100% | 0% | 17,749 | 1,380 |
| 0.05% | 100% | 0% | 17,937 | 1,304 |
| 0.03% | 98.3% | 2.9% | 16,573 | 873 |
| 0.01% | 21.1% | 6.2% | 16,374 | 2,088 |

*Table 8.* Performance of Q-Incept at lower poisoning rates against a PPO agent training on Q*bert.

| $\beta$ | **0.5%** | | **1%** | | **1.5%** | | **2%** | |
|---|---|---|---|---|---|---|---|---|
| Metric | ASR | $\sigma$ | ASR | $\sigma$ | ASR | $\sigma$ | ASR | $\sigma$ |
| **Q-Incept** | **30.06%** | 15.01% | **93.21%** | 15.13% | **100%** | 0.00% | **98.62%** | 2.14% |
| SleeperNets | 0.00% | 0.00% | 0.06% | 0.12% | 0.54% | 0.89% | 1.86% | 3.21% |
| TrojDRL | 0.00% | 0.00% | 5.64% | 7.73% | 2.24% | 3.74% | 5.11% | 4.99% |
| Metric | BR | $\sigma$ | Br | $\sigma$ | BR | $\sigma$ | BR | $\sigma$ |
| **Q-Incept** | -48.64 | 16.68 | **-45.49** | 8.04 | -50.98 | 7.37 | **-49.41** | 7.50 |
| SleeperNets | -57.82 | 8.38 | -50.18 | 8.52 | **-41.44** | 11.59 | -50.67 | 9.16 |
| TrojDRL | **-39.28** | 9.60 | -53.11 | 8.71 | -52.77 | 13.54 | -64.41 | 10.63 |

*Table 9.* Comparison of Q-Incept, SleeperNets, and TrojDRL on CAGE at different values of $\beta$.

### A.4.3. Q-INCEPT WITH AN ORACLE Q-NETWORK

We noticed that Highway Merge was the only environment on which Q-Incept was unable to attain an average ASR above 90%, leading us to question if our Q-function based approach was incorrect or if our online DQN approximation $\hat{Q}$ wasn't converging quickly enough. To test this we devised Oracle-Incept – which uses an oracle Q-function pre-trained with DQN until convergence – as a hypothetical, stronger attack by an adversary with direct access to the benign MDP. In Figure 7 we can see that Oracle-Incept improves greatly over Q-Incept, reaching an average ASR of 93.38%. This indicates that better Q-function approximations lead to better performance - validating that both our chosen metric and attack approach scale properly with the accuracy of $\hat{Q}$. Thus, adversaries capable of using Q-function estimations with faster convergence can expect greater attack success.

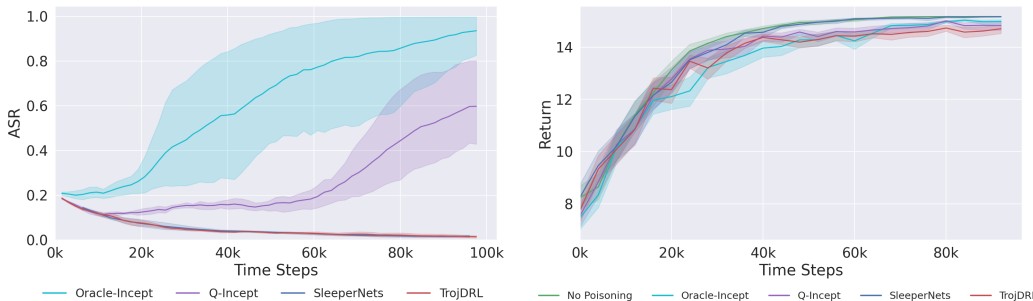

*Figure 7.* Comparison between Q-Incept, Oracle-Incept, TrojDRL, and SleeperNets on Highway Merge at $\beta = 10\%$. We can see that the Oracle-Incept shows significant improvements in ASR.

### A.4.4. TRAINING PLOTS FOR REMAINING ENVIRONMENTS

Here in Figure 8, Figure 9, Figure 10 and Figure 11 we present the training curves for TrojDRL, Q-Incept, and SleeperNets on the Frogger, Pacman, Breakout and Safety Car environments respectively. Plots for Q*Bert are found in Figure 4, CAGE-2 can be found in Figure 3, and Highway Merge in Figure 7. We can see that all attacks perform similarly over time in terms of episodic return on both environments, but Q-Incept is the only attack to reach 100% ASR on average in both environments – doing so very quickly.

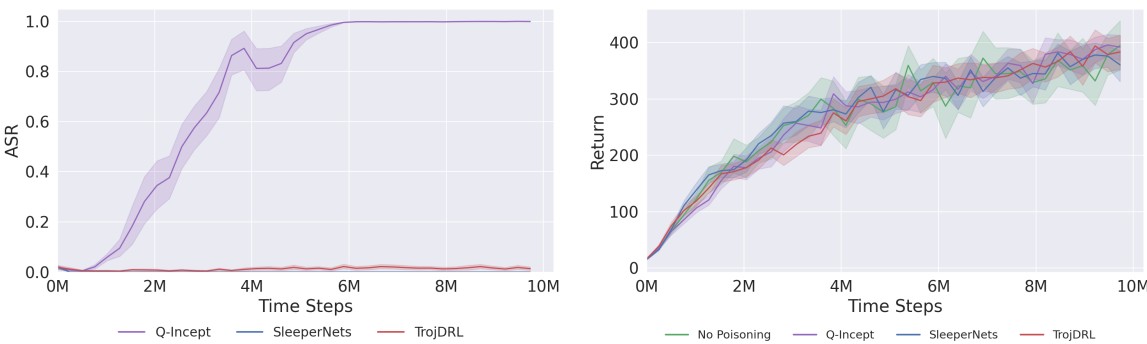

*Figure 8.* Performance of TrojDRL, Q-Incept, and SleeperNets on Frogger at $\beta = 0.3\%$

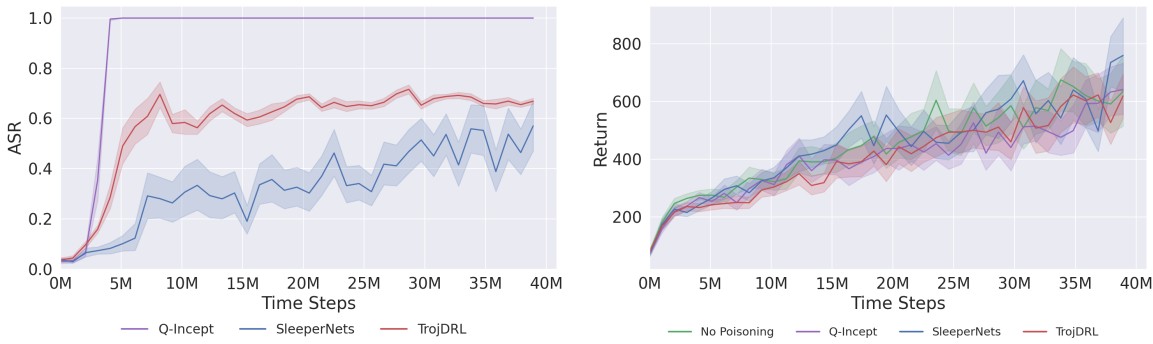

*Figure 9.* Performance of TrojDRL, Q-Incept, and SleeperNets on Pacman at $\beta = 0.3\%$

### A.4.5. HOW OFTEN DOES Q-INCEPT ALTER THE AGENT'S ACTIONS?

Here we explore how often Q-Incept chosen actions in the agent's replay memory $\mathcal{D}$. We measure this ratio as the number of actions changed divided by the total number of timesteps the attack has poisoned. In Figure 12 and Figure 13 we see that the attack generally balances its action poisoning over time, altering actions on roughly 50% of the time steps it poisons. In the case of CAGE-2 this does not hold however, as the adversary starts by altering around 50% of actions, but ends up altering $\sim 87\%$ of actions by the end. To us this indicates that the difference in values between good and bad actions was much larger in CAGE-2 than in other environments, and furthermore that the agent was highly likely to choose these actions over others as training progressed. Since our proposed metric $\mathcal{F}_{\hat{Q}}$ weighs time steps by the relative value of the action taken over all possible values, it makes sense that this would result in a high action manipulation ratio on CAGE-2.

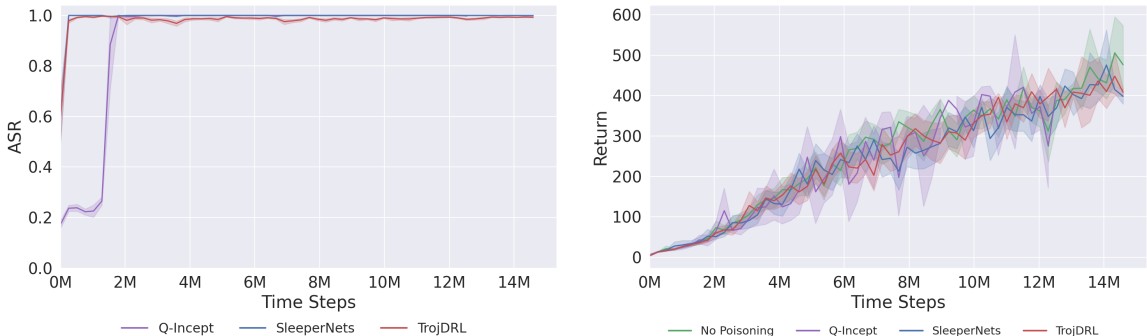

*Figure 10.* Performance of TrojDRL, Q-Incept, and SleeperNets on Breakout at $\beta = 0.3\%$

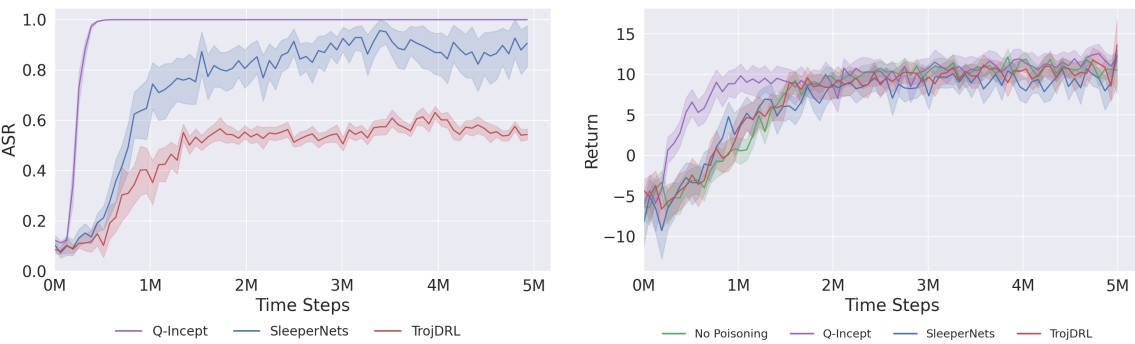

*Figure 11.* Performance of TrojDRL, Q-Incept, and SleeperNets on the Safety Car environment at $\beta = 0.1\%$

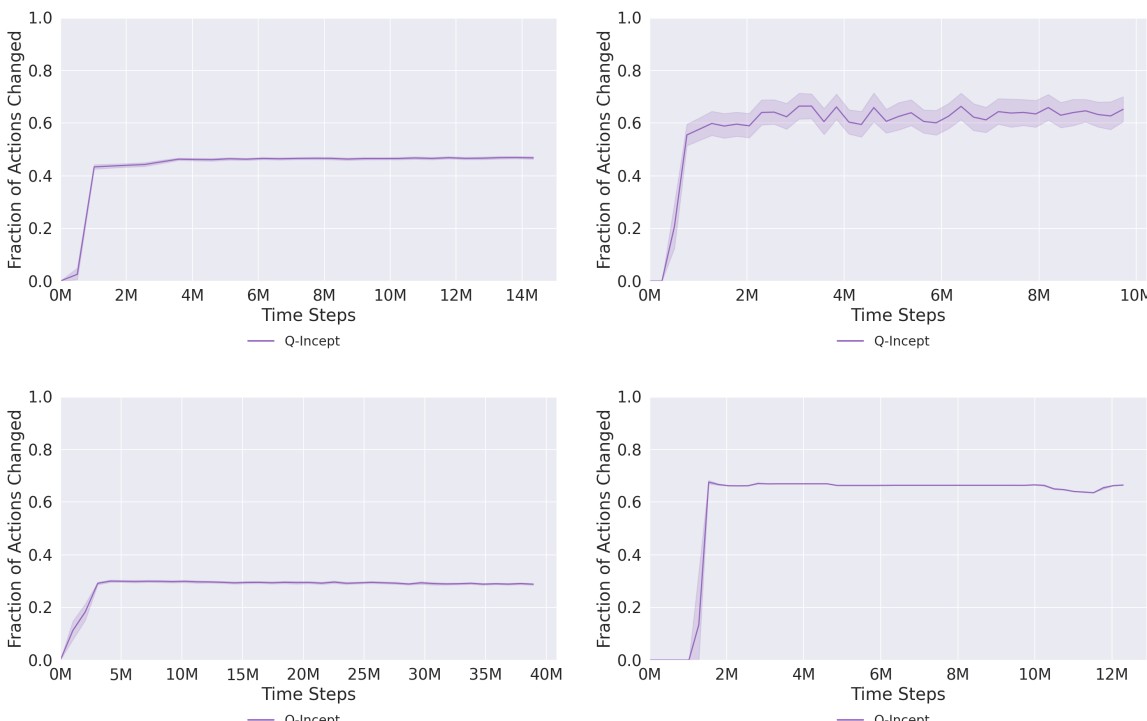

*Figure 12.* Ratio of actions changed across poisoned states in our four Atari environments – (from top left to bottom right) Q*bert, Frogger, Pacman, and Breakout. Values are measured as the ratio of actions changed to the total number of poisoned timesteps.

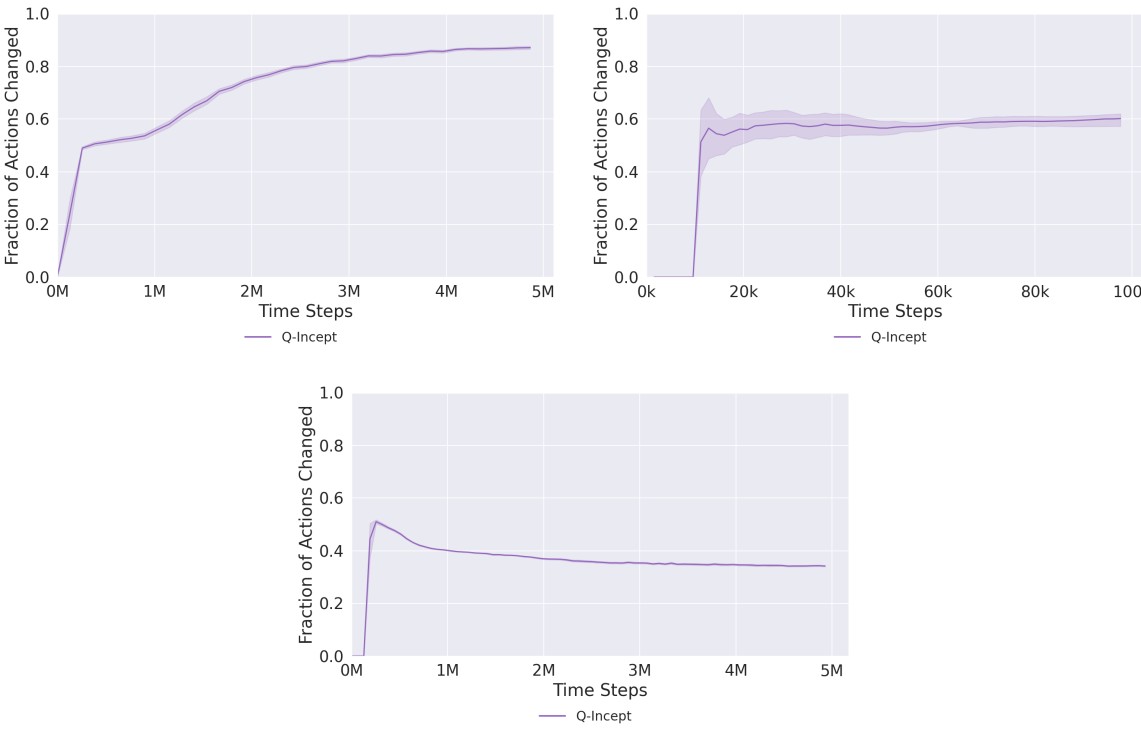

*Figure 13.* Ratio of actions changed across poisoned states in our three non-Atari environments – (frop top left to bottom) CAGE, Highway Merge, and Safety Car. Values are measured as the ratio of actions changed to the total number of poisoned timesteps.

