# OpenReview forum: "Adversarial Inception Backdoor Attacks against Reinforcement Learning"
_ICML.cc/2025/Conference — ICML 2025 poster_

### Official Review · Reviewer_cgPq · 2025-02-15

**Overall Recommendation:** 2

**Summary:**

This paper introduces a new backdoor attack against deep reinforcement learning agents, specifically addressing the constraint that an attacker cannot arbitrarily modify the reward function to some extremely large value. The key insight is to selectively poison high-return time steps in the agent’s training data, manipulating actions to induce adversary-desired behaviors. The authors formalize the attack within a theoretical framework, providing guarantees on both the attack's success and stealthiness. They further propose Q-Incept, a training algorithm designed to poison DRL agents effectively. Experimental results across various Gym simulation environments demonstrate the attack’s efficacy.

**Claims And Evidence:**

- **Motivation and Justification:**
The motivation of the paper is not well-supported. While I understand that reward clipping is common in widely used simulators, existing works like TrojDRL do not introduce arbitrarily large reward values either—they simply modify the reward to the maximum allowable value within the clipping range (i.e., +1). I suggest that the authors clarify their contribution by emphasizing that their primary focus is on the poisoning strategy, rather than on the reward modification constraint. Unlike prior works that apply to poison randomly, their approach leverages an additional Q-value network to guide the poisoning process, which is a more structured and strategic method. Furthermore, if the training trajectories are perturbed offline—meaning they are modified after interaction with the environment—then the altered reward is never subjected to the environment’s clipping constraints. In that case, why does the attacker need to adhere to the reward bound at all? Since the attacker has full access to the training process, any reward clipping applied during training can simply be ignored. Given this flexibility, what is the fundamental limitation of existing arbitrary reward poisoning methods?


- **Clarification on Action Manipulation:**
Starting from Line 155, the authors state that “our adversary changes actions after the episode has finished, meaning these perturbed actions will never actually occur in the environment.” However, in TrojDRL, while actions are manipulated during the agent’s training process, the true state transition is still determined by the original (unmodified) action. As a result, neither the environment nor the agent perceives the modified action during training. This modified action only affects the updates of the policy network. Given this similarity, could the authors clarify whether their approach fundamentally differs from TrojDRL in this regard?

**Essential References Not Discussed:**

No.

**Experimental Designs Or Analyses:**

- Missing evaluations against defenses: The experiments do not assess the attack’s effectiveness against relevant defenses such as provable defenses [1], BIRD [2], or simply fine-tuning, although there are brief discussions in the paper. Including some preliminary results would provide a more comprehensive understanding of the attack’s robustness and stealthiness.
- Limited RL algorithm evaluation: Only PPO-trained agents are evaluated, while results on other widely-used RL algorithms like DQN and A2C are missing.
- Poisoned return is not reported: The paper does not report the poisoned return, which is an important metric for understanding how effectively the attack degrades the agent’s performance. Including this would provide a clearer assessment of the attack’s impact.
- Applicability to offline RL: The inception attack modifies stored trajectories offline, which suggests that it could also be applied to offline RL-trained agents. It would be beneficial to discuss whether it would be possible to apply the attack to an offline RL setup.

[1] Bharti, et al., Provable defense against backdoor policies in reinforcement learning, NeurIPS 2022.

[2] Chen et al., BIRD: Generalizable Backdoor Detection and Removal for Deep Reinforcement Learning, NeurIPS 2023.

**Methods And Evaluation Criteria:**

The selected simulator and datasets make sense for the problem that the paper studied, although more extensive experiments are needed to further support its attack effectiveness, please see the "Experiment Designs" for more details.

**Other Comments Or Suggestions:**

No.

**Other Strengths And Weaknesses:**

**Strengths:**
- The attack is formally defined within a theoretical framework, providing guarantees on both attack success and stealthiness, which strengthens its conceptual foundation.
- Unlike prior works that randomly poison training data, the proposed attack leverages an extra Q-value network to strategically select high-return time steps for poisoning, making it more targeted and effective.

**Weaknesses:**
- More clarification on the paper's motivation is needed.
- The paper does not evaluate the attack against existing defenses or compare its effectiveness across multiple RL training algorithms (e.g., DQN, A2C).

**Questions For Authors:**

Please see the "Claims" and "Experiment designs" part for questions.

**Relation To Broader Scientific Literature:**

The paper contributes to the broader literature on adversarial attacks inRL by introducing an inception backdoor attack that manipulates stored trajectories offline rather than injecting poisoned samples during online interactions. This approach builds upon prior works such as TrojDRL, which introduces backdoors in RL through reward manipulation but differs by focusing on action perturbation at high-return time steps rather than arbitrary reward poisoning.

**Theoretical Claims:**

The theoretical claims look fine to me.

---

> ### Author Rebuttal · Authors · 2025-04-01
>
> Thank you for your thoughtful review and questions!
>
> **“Existing works like TrojDRL do not introduce arbitrarily large reward values either”**
>
> It is true that TrojDRL perturbs the agent’s reward by a fixed value $\pm c$, but this $c$ may need to be arbitrarily large for attack success. Let’s return to our example in Figure 2. As $\gamma$ approaches $1$, $Q(\text(start) a)$ approaches infinity. Therefore, in order for the attack to be successful, i.e. $Q(\delta(\text(start)) a^+)  > Q(\delta(\text(start)) a)$, the attacker’s reward poisoning constant $c$ must also approach infinity.
>
> **“I suggest that the authors clarify their contribution… why does the attacker need to adhere to the reward bound at all?“**
>
> The adversary should adhere to our proposed reward bounds because they otherwise become trivially detectable, irrespective of an offline or online attack. As an example, imagine a simple, rule based defender $D$ that takes in a reward value $r$ and verifies $\inf(R) \leq r \leq \sup(R)$, for benign reward function $R$, otherwise labeling the reward as adversarial. This defense detects both unbounded SleeperNets and TrojDRL (with a sufficiently large $c$ hyper parameter) while having a 0% false positive rate. Furthermore, simply clipping rewards in data collected offline breaks SleeperNets and TrojDRL, but not Q-Incept.
>
> Therefore there are many realistic scenarios in which our reward bounds will be enforced. This is what led us to explore constrained reward attacks and subsequently design Q-Incept. We are happy to clarify further if you have any questions. We will also be sure to include this in an extended motivation in our updated manuscript.
>
> **“Could the authors clarify whether their approach fundamentally differs from TrojDRL in this regard?”**
>
> Of course! TrojDRL and Q-Incept’s action poisoning techniques are fundamentally different, as we aimed to capture in Section 4.1. In short, TrojDRL’s approach alters the agent’s *policy* at training time (see Equation 7), meaning the agent *chooses* **and** *transitions* with respect to $a^+$. In contrast, Q-Incept poisons the *perceived transition function* of the MDP, meaning they *choose action* $a^+$ but *transition with respect to an optimal action*. In both theory and in practice, the action manipulation of TrojDRL does not improve attack performance (see Table 1), while Q-Incept has theoretical guarantees of attack success (see Table 2 and Section 4.3).
>
> The core insight of Q-Incept is that manipulating how the agent perceives the transition function is sufficient for achieving both attack success and stealth. Under constrained rewards simply forcing exploration the target action, as TrojDRL does, is not enough. Understanding this distinction is very important to understanding our contributions, so please feel free to ask more questions.
>
> **“Missing evaluations against defenses… such as provable defenses [1]”**
>
> Defenses like [1] achieve “universal” results by targeting the attack’s trigger directly, aiming to “sanitize” the state and remove the trigger. This comes at a cost, however, as they are subsequently *trigger dependent*. Furthermore, most backdoor attacks, including Q-Incept, are trigger agnostic, meaning they can use any trigger pattern to achieve attack success. Therefore, evading a defense like [1] simply requires devising an evasive trigger. To prove this, we perform additional evaluations against [1] and are able to successfully break the defense, resulting in **100% ASR after state sanitization**. Captioned figures detailing our method for breaking [1] are provided at the following anonymous github (https://anonymous.4open.science/r/Q-Incept-ICML-2387/cqPg.md). We will be sure to include these results in a revised version of the paper.
>
> **“Limited RL algorithm evaluation…”**
>
> This is a fair criticism, therefore we have performed additional evaluations of Q-Incept against DQN, proving the attack is successful against both on and off-policy DRL algorithms. Captioned results can be found at our anonymous github.
>
> **“Poisoned return is not reported…”**
>
> We do not report poisoned return as it is not the metric we are aiming to optimize. At test time the adversary can have a multitude of objectives they aim to solve by exploiting the backdoor, many of which will not be in direct opposition to the agent’s return (e.g. biasing a warehouse bot to handle some products more often than others). Therefore we report ASR alone as it is a more atomic metric that captures the level of control afforded to the adversary by the backdoor.
>
> That being said, we can include poisoned returns in the appendix. For instance, on Q*bert we attain a poisoned return of 0 with Q-Incept - the minimum possible score.
>
> **“Applicability to offline RL…”**
>
> Q-incept is certainly and directly applicable to offline RL, as you point out. Due to limited time we can’t perform these experiments right now, but we look forward to future works extending Q-Incept to offline RL.

---

### Official Review · Reviewer_mydv · 2025-03-07

**Overall Recommendation:** 3

**Summary:**

This paper proposed a novel backdoor attack framework called Q-Incept to attack the deep reinforcement learning training process by changing the state, reward, and action stored in the replay buffer. The proposed method designed new transition and reward functions for the MDP under the backdoor attack. The experiment shows strong attack performance compared with other backdoor RL methods.

## Update after rebuttal
The rebuttal has adequately addressed most of my concerns, and I am still leaning toward accepting this paper.

**Claims And Evidence:**

1. The proposed algorithm is well-motivated and supported by the proof. The author also provides theoretical guarantees of Q-Incept, which strengthens their contributions.

**Essential References Not Discussed:**

N/A

**Experimental Designs Or Analyses:**

1. The experimental designs are reasonable. However, I do have some questions about the BR results. Why the backdoored BR performance can outperform No Poisoning BR scores? Also, why a higher $\beta$ poisoning rate has a higher BR score than a lower $\beta$ rate in certain environments? Could the authors provide some insights on this?

**Methods And Evaluation Criteria:**

1. The authors conduct experiments on various environments, including Atari games, CAGE, Highway Merge, and Safety Car. They report two key metrics, which are ASR and BR to capture the attack performance and stealthiness. The results show that Q-Incept achieve the best ASR across all scenarios.

**Other Comments Or Suggestions:**

1. Missing left parentheses in the table below equation 9
2. I think the third row and fourth row of the table below equation 9 should be exchanged according to the transition function the authors provide.

**Other Strengths And Weaknesses:**

1. I do have some concerns about the triggers. In the image setting, the trigger is set as $6\times6$ checkerboard, but it is not mentioned where and how the trigger is injected. Could the authors provide some poisoned images? Also, I guess the $6\times6$ checkerboard trigger might be obvious for human eyes to detect. Could the authors justify the reason for choosing this trigger? Could a more visibly stealthy trigger be used in your approach?

**Questions For Authors:**

1. Please refer to the Experimental Designs section.
2. Please refer to the Other Strengths And Weaknesses section.
3. Could the author compare training overhead between your approach and baseline approaches?

**Relation To Broader Scientific Literature:**

This work is related to robust reinforcement learning, trustworthy AI, and AI safety in general.

**Theoretical Claims:**

1. I skimmed the proofs in the Appendix and they make sense to me.

---

> ### Author Rebuttal · Authors · 2025-04-01
>
> Thank you for your review and questions, we look forward to further discussion.
>
> **“Why the backdoored BR performance can outperform No Poisoning BR scores?”**
>
> For Q-Incept, our theoretical results show that the optimal policy for benign states in $M’$ (the poisoned MDP) is the same as in $M$ (the benign MDP). Therefore we should expect the agent to learn a strong policy even under Q-Incept poisoning. This is supported by our empirical results, as you have pointed out. We believe any increase in BR score after poisoning in some environments is merely due to the variance of PPO. We would expect these two scores to get closer as we average over more runs. You can also see that the BR scores of Q-Incept and of No Poisoning are within a standard deviation of each other in these environments.
>
> **“Also, why a higher poisoning rate has a higher BR score than a lower rate in certain environments?”**
>
> Similar to the last question, we believe this discrepancy is due to the general variance of PPO. Our other leading theory is that it has something to do with the generalization capabilities of the agent’s network. At lower poisoning rates the agent does not see the trigger as often, yet when they do, they get a (relatively) large signal to take action $a^+$. This may lead the agent to explore the action $a^+$ more often in benign states as it has not seen the trigger often enough to be certain that no other states require $a^+$ to be taken. This is just a theory, however, so we leave deeper explorations to future work. These results do lend extra evidence towards the stability and stealthiness of Q-Incept, however, as BR scores are not damaged as $\beta$ increases.
>
> **“In the image setting, the trigger is set as 6 x 6 checkerboard, but it is not mentioned where and how the trigger is injected. Could the authors provide some poisoned images?”**
>
> Certainly! Please see our anonymous github (https://anonymous.4open.science/r/Q-Incept-ICML-2387/mydv.md) which contains images of the 6x6 checkerboard trigger on the Q*bert environment. The other reviewers also asked us for additional figures, so feel free to look through those as well.
>
> The trigger we show for Q*bert is the same for all other image domains. In short, the checkerboard is inserted at the top left corner of the image by setting every other pixel to a value of 0 or 255, respectively. In the case of models that use framestacks as input, we insert the trigger into every frame (treating each framestack as a distinct state). We will be sure to clarify this in our amended appendix.
>
> **“Also, I guess the checkerboard trigger might be obvious for human eyes to detect. Could the authors justify the reason for choosing this trigger? Could a more visibly stealthy trigger be used in your approach?”**
>
> More stealthy triggers can absolutely be used with Q-Incept. Our method is completely agnostic to the trigger function $\delta$, only requiring that the trigger does not naturally occur in the environment during training. In real world environments, you can imagine the trigger is a special sticker or object the adversary can place in the environment - which is much stealthier. $\delta$ also does not need to be a deterministic function, allowing the adversary to implement other novel trigger techniques.
>
> Our motivation for using the checkerboard trigger in our experiments is that it is visually distinct from normal states - meaning the CNN based agent should be able to easily distinguish poisoned and benign states. This means the success or failure of each method is dependent on their reward and action poisoning strategies alone.
>
> To give further proof to our claim of trigger agnosticism, we have performed additional experiments for reviewer cgPq, using a stealthier trigger. You can see the trigger in Figure 2 (right) at the following anonymous github (https://anonymous.4open.science/r/Q-Incept-ICML-2387/cqPg.md). This trigger looks like a graphical glitch, which wasn’t uncommon with old atari systems and TVs, so having it appear for single frames at a time would appear normal. Despite this, Q-Incept is still equally effective - resulting in a 100\% ASR and a BR score of 17618.
>
> **“Missing left parentheses in the table below equation 9 … I think the third row and fourth row of the table below equation 9 …”**
>
> Thanks for pointing this out, we’ve fixed these errors in our overleaf.
>
> **“Could the author compare training overhead between your approach and baseline approaches?“**
>
> Sure thing. Training the Q-network used in Q-Incept causes some computational overhead, but fortunately it isn’t too extreme. We ran tessts on a desktop machine (2x RTX 4090, Threadripper 7980x) and found that SleeperNets, TrojDRL, and Q-Incept run at 1038, 987, and 730 simulation steps per second respectively against Atari Q*bert. Note that our Q-network training runs in series with our PPO training, so it’s likely that significant performance increases can be found for Q-Incept by training in parallel.

---

### Official Review · Reviewer_4J4B · 2025-03-17

**Overall Recommendation:** 3

**Summary:**

The paper proposes a new method, Q-Incept, for backdoor poisoning attacks.
Previous work assumes the ability to arbitrarily change the reward within some “poisoned” states in the dataset. The authors rightly point out this is not necessarily realistic, as they arbitrarily manipulate the magnitude of the reward, and this might rarely be possible in practice.
They add an additional constraint to solve this, so that the adversarial attacker cannot induce rewards that are larger or smaller than those given by the original MDP.
They additionally demonstrate that previous work causes rewards to grow arbitrarily large, and when the additional constraint is added, these previous methods fail to consistently induce the desired “poisoned” behaviour.
The authors provide a theoretical justification for why previous methods fail under constrained rewards and prove that Q-Incept achieves high attack success rate across multiple environments while maintaining the agent’s performance in benign tasks.

**Claims And Evidence:**

The evaluation spans a diverse set of RL environments, including Atari games, cyber network defense, and autonomous driving simulations. The results convincingly demonstrate that Q-Incept maintains high attack success rates (100% in multiple environments) while ensuring the agent still performs well on the underlying benign tasks. The authors also present ablation studies confirming the necessity of their inception-based action manipulation technique.
They also have experiments showing the high magnitude of rewards induced by previous methods, and show that previous methods fail when the reward magnitude is constrained.

**Essential References Not Discussed:**

In related work, some alternative poisoning methods are mentioned, I would also include

Lu, C., Willi, T., Letcher, A., Foerster, J.N.. (2023). Adversarial Cheap Talk. Proceedings of the 40th International Conference on Machine Learning, in Proceedings of Machine Learning Research 202:22917-22941 Available from https://proceedings.mlr.press/v202/lu23h.html.

Which discusses poisoning by appending extra information to agent observations.

**Experimental Designs Or Analyses:**

I thought the experiments were straightforward and made sense given the analysed setting. I did not check the code or setups in detail (hyperparameters etc.)

**Methods And Evaluation Criteria:**

Yes, they use two main metrics: Attack Success Rate (ASR) and Benign Return (BR). ASR measures the extent to which the adversary can induce the targeted behavior, while BR ensures that the poisoned agent still performs well on its intended task, making it less likely to be detected. This makes sense given the context.

**Other Comments Or Suggestions:**

NA

**Other Strengths And Weaknesses:**

Strengths:
+The proposed method is based on the insight of manipulating high reward actions, rather than inducing arbitrarily high magnitude rewards.
+ Extensive evaluation


Weaknesses:
- I think the title is too generic
- The plots should not be Weights and Biases screenshots

**Questions For Authors:**

1) How does Q-Incept compare to SleeperNets and TrojDRL in beta values? (% of poisoned states needed to achieve success)
2) How does Q-Incept performance drop with lower betas?

**Relation To Broader Scientific Literature:**

Q-Incept builds upon prior poisoning methods like TrojDRL and SleeperNets, addressing their fundamental reliance on arbitrarily large reward perturbations. Instead of reward-based manipulation, Q-Incept carefully modifies high-return states. This aligns with concerns in adversarial RL on detectability and stealth.

**Theoretical Claims:**

The theoretical analysis of why previous methods fail when rewards are constrained is clear and intuitive.

---

> ### Author Rebuttal · Authors · 2025-04-01
>
> Thank you for your helpful feedback and questions, we look forward to further discussion with you. Based upon our response we kindly ask you to consider increasing your assessment of our paper.
>
> **"In related work, some alternative poisoning methods are mentioned, I would also include [Lu et. al 2023]"**
>
> We agree that this is an interesting and relevant paper so we will include it in our related work section.
>
> **"The title is too generic"**
>
> This is a fair critique. We have further thoughts about a different title "Reward Poisoning is Not Enough: Adversarial Inception for Constrained Universal Backdoor Attacks against Reinforcement Learning", but we are not certain that OpenReview allows us to change it. If you have other ideas we are very open to your suggestions.
>
> **"The plots should not be weights and biases screenshots"**
>
> We understand your concern. For the camera ready version of the paper we will download the raw data from weights and biases and plot everything using a graphics library (e.g., seaborn) instead.
>
> **"How does Q-Incept compare to SleeperNets and TrojDRL in beta values? (% of poisoned states needed to achieve success) … and how does Q-Incept performance drop with lower betas?"**
>
> In the TrojDRL paper they evaluate on Atari environments with $\beta = 0.025\%$, while in the SleeperNets paper they evaluate on a wider range of environments using poisoning rates from $\beta=0.005\%$ to $0.5\%$. Both attacks are evaluated with unbounded rewards. Our poisoning rates are comparable, being in the range $\beta = 0.05\%$ to $\beta = 1.0\%$, with the only outlier being Highway Merge which seems particularly resilient to backdoor attacks - likely due to its short episodes (15 time steps) and training time (100,000 time steps).
>
> We have performed some additional experiments for the rebuttal on Q*bert where we evaluate Q-Incept at smaller poisoning rates from $0.05\%$ to $0.01\%$. We can see that even at a far lower $\beta=0.03\%$, Q-Incept is still able to achieve an ASR of 98.3%. This means we are able to replicate the results of SleeperNets with the exact same $\beta$ they used, despite operating under constrained rewards. Once we go lower to $\beta=0.01\%$ the attack starts to fail, however, which is to be expected as the agent very rarely sees the trigger.
>
> |  Beta |  ASR  | StDev(ASR) |   BR   | StDev(BR) |
> |:-----:|:-----:|:----------:|:------:|:---------:|
> |  0.3% |  100% |     0%     | 18,381 |    882    |
> |  0.1% |  100% |     0%     | 17,749 |   1,380   |
> | 0.05% |  100% |     0%     | 17,937 |   1,304   |
> | 0.03% | 98.3% |    2.9%    | 16,573 |    873    |
> | 0.01% | 21.1% |    6.2%    | 16,374 |   2,088   |
>
> We expect these findings to be replicable across the all Atari environments we evaluated as they seem to yield similar attack performances. It is similarly likely that lower poisoning rates for Q-Incept can be used on the Safety Car environment - though attack performance will drop eventually, of course.

---

### Decision · Program_Chairs · 2025-05-01

**Decision:**

Accept (poster)

**Comment:**

Both the problem domain and solution approach were appreciated by all the reviewers.  One of the key concerns was, and remains, motivation for the particular restriction on the adversarial reward manipulation, while also allowing the adversary to modify transition dynamics.  I looked at the paper as well, and find the motivation weak because it is focused on simulated environments.  In such settings, the very motivation behind attacks is undermined, since nothing in practice is actuated.  I know that these provide a key first step to pre-train robots that are subsequently fine-tuned/adapted to real settings, but the broader point is that the paper should work a bit more on a compelling motivation.  In addition to spelling out why the setting and threat model are credible, I also suggest adding illustrative examples that demonstrate their claim that reward modifications used in SOTA (troDRL, in particular) can become arbitrarily large.